# REAP THE EXPERTS: WHY PRUNING PREVAILS FOR ONE-SHOT MoE COMPRESSION

**Mike Lasby**[1,2,†], **Ivan Lazarevich**[1], **Nish Sinnadurai**[1], **Sean Lie**[1],
**Yani Ioannou**[2], **Vithursan Thangarasa**[1]
[1]Cerebras Systems Inc., [2]Schulich School of Engineering, University of Calgary

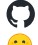 https://github.com/CerebrasResearch/reap
https://hf.co/collections/cerebras/cerebras-reap

## ABSTRACT

Sparsely-activated Mixture-of-Experts (SMoE) models offer efficient pre-training and low latency but their large parameter counts create significant memory overhead, motivating research into expert compression. Contrary to recent findings favouring expert *merging* on discriminative benchmarks, we find that expert *pruning* is a superior strategy for generative tasks. We demonstrate that existing merging techniques introduce an irreducible error due to the loss of fine-grained routing control over experts. Leveraging this insight, we propose Router-weighted Expert Activation Pruning (REAP), a novel pruning criterion that considers both router gate-values and expert activation norms to minimize the reconstruction error bound. Across a diverse set of SMoE models ranging from 20B to 1T parameters, REAP consistently outperforms merging and other pruning methods on generative benchmarks, especially at 50% compression. Notably, our method achieves near-lossless compression on code generation tasks with Qwen3-Coder-480B and Kimi-K2, even after pruning 50% of experts.

## 1 INTRODUCTION

Interest in the Sparsely-activated Mixture-of-Experts (SMoE) architecture for Large Language Models (LLMs) surged following the release of DeepSeek-V3 (DeepSeek-AI et al., 2024) and other high-quality open-weight SMoE LLMs (Jiang et al., 2024; Meta AI Team, 2025; Yang et al., 2025a; Zeng et al., 2025; Baidu, 2025; Kimi Team et al., 2025). Compared to dense models, SMoEs offer lower latency and more efficient pre-training (Fedus et al., 2022). However, SMoEs require more parameters than dense models to achieve similar accuracy, resulting in significant memory overhead. Further, expert usage imbalance during inference causes poor accelerator utilization, leading to increased latency or compromises such as dropped tokens (Balmau et al., 2025). Expert usage imbalance also represents an opportunity, motivating prior work which investigates whether experts can be compressed without negatively impairing accuracy (Li et al., 2023; Lu et al., 2024). By eliminating or compressing redundant experts, memory overhead is reduced. A more uniform distribution of expert usage would also improve hardware utilization. Expert compression is particularly valuable for use cases which feature small batch sizes such as local deployments and academic research.

Initial expert compression efforts focused on expert pruning, the removal of experts in their entirety. However, expert pruning is a strong intervention on the model's weights. Techniques such as quantization, low-rank compression, and expert merging also offer memory savings but maintain a lossy representation of the less important experts. Crucially, expert merging has recently been demonstrated to outperform expert pruning when evaluated with perplexity and on Multiple Choice (MC) question answering benchmarks (Li et al., 2023; Liu et al., 2024b). However, an evaluation comparing these methods on generative benchmarks has yet to be conducted. In this work, we demonstrate that — when paired with a suitable saliency criterion — expert pruning outperforms expert merging, particularly on generative benchmark tasks such as code generation, creative writing, and mathematical reasoning. Specifically, our main contributions are as follows:

- We demonstrate that existing expert merging techniques introduce *irreducible error* due to the loss of the router's independent, input-dependent modulation of the expert outputs. In *high-granularity* SMoEs, the loss of fine-grained routing control results in *functional subspace collapse*;
- Empirically, we find that expert merging distorts the functional manifold topology due to the introduction of novel functionality. Conversely, as a coordinate subspace operation, pruning preserves the topology;

---

Correspondence to mklasby@ucalgary.ca & vithu@cerebras.net
† Work completed while on internship at Cerebras

- We introduce Router-weighted Expert Activation Pruning (REAP), a novel expert pruning saliency criterion. By considering both router gate-values and expert activation norms, REAP explicitly minimizes the upper bound of the reconstruction error derived in our analysis by pruning experts which contribute minimally to the layer output;
- Across diverse SMoE architectures ranging from 20B to 1T parameters and a suite of generative evaluations, we demonstrate the significant and consistent advantage of REAP over existing expert pruning and merging approaches, particularly at 50% compression. Notably, our method achieves near-lossless compression ($\Delta_{acc} \leq 2\%$) on code generation tasks after pruning 50% of experts from Qwen3-Coder-480B and Kimi-K2;
- We open-source our code and select compressed model checkpoints to facilitate further research on compressed SMoEs and their applications.

## 2 RELATED WORK

**Sparsely activated SMoE architecture.** A Mixture-of-Experts (MoE) layer is comprised of multiple, specialized feed-forward subnetworks known as *experts* and a router which produces gate-values (i.e., *gates*) to dynamically modulate the output of the experts based on the input. The architecture was revived in the deep learning era by the introduction of the SMoE by Shazeer et al. (2017). SMoEs layers only select a subset of experts to use for each input, enabling massive scaling of model parameters without a commensurate increase in computational cost (Lepikhin et al., 2021; Fedus et al., 2022). In transformer-based LLMs, SMoE layers are integrated by replacing the traditional feed-forward layers. Further innovations such as auxiliary-loss-free load balancing (DeepSeek-AI et al., 2024), shared experts, and fine-grained experts (Dai et al., 2024) have propelled SMoE architectures to become the *de facto* standard for LLMs in recent months.

**Expert pruning.** Although SMoE layers effectively decouple total model parameters from inference costs, their memory overhead has motivated research in expert pruning to reduce total number of parameters. Early efforts demonstrated that progressively pruning experts based on router weights during fine-tuning until a single expert remained could preserve model quality in task-specific settings (Chen et al., 2022). Koishekenov et al. (2023) found expert pruning to be effective without further fine-tuning despite aggressively pruning up to 80% of experts. Muzio et al. (2024) found that global pruning using gate-values as a saliency criterion was more effective than uniform, layer-wise frequency-based pruning. Other sophisticated pruning criteria have been proposed: Lu et al. (2024) introduced an exhaustive search strategy which prunes experts that minimize the reconstruction loss between the original and pruned layer outputs; Liu et al. (2024a) used a gradient-free evolutionary algorithm to prune experts. Both of these works demonstrated significant improvements over naive frequency-based pruning. A comprehensive evaluation of 16 diverse pruning criteria was conducted by Jaiswal et al. (2025). Expert Activation Norm (EAN) was empirically found to be the highest performing criterion and the benefits of iterative pruning were presented. EASY-EP (Dong et al., 2026) proposed expert pruning based on the sum of gate-weighted expert output norms. In contrast, REAP ensures a frequency-agnostic assessment by computing the conditional mean strictly over the tokens where an expert is active.

**Expert merging.** While the above-noted works prove that expert compression is feasible via pruning, an alternative compression technique is to *merge* experts. Generally, merging requires both a clustering algorithm and a merging technique. Li et al. (2023) introduced Merge Sparse Mixture of Experts (M-SMoE) which first initializes expert cluster centres by identifying the *dominant* experts with the highest usage frequency globally across all layers. The remaining non-dominant experts are clustered based on the cosine similarity of router logits. Finally, expert weights are aligned via permutation with the weight matching algorithm (Ainsworth et al., 2023) and merged using frequency-weighted parameter averaging. Li et al. (2023) found that their technique outperformed Chen et al.'s (2022) pruning method on MC benchmarks. Chen et al. (2025) proposed Hierarchical Clustering for Sparsely activated Mixture of Experts (HC-SMoE). HC-SMoE clusters experts based on the euclidean similarity of their *representative vectors* — the average activation of each expert measured on *every* token in a calibration dataset — using hierarchical agglomerative clustering. Similar to M-SMoE, HC-SMoE uses frequency-weighted parameter averaging to merge clusters into a single merged expert. Without any fine-tuning, Chen et al. (2025) found that their technique outperformed expert pruning based on router logits (He et al., 2025a), frequency, and Lu et al.'s (2024) method when benchmarked on a suite of MC question answering tasks.

**Other compression techniques.** In addition to pruning and merging, experts may be compressed through quantization (Huang et al., 2025; Li et al., 2025; Duanmu et al., 2025), low-rank decomposition (Yang et al., 2024a; Gu et al., 2025; He et al., 2025b), weight sparsity (He et al., 2025a), or a combination of

any of the above techniques (Liu et al., 2025a). These other approaches are orthogonal to expert pruning and merging; however, note that expert merging necessitates re-quantization for block quantization formats that share common scaling coefficients across a group of weights whereas pruning does not.

**Model merging.** Model merging aims to combine parameters from multiple trained neural networks and has been rapidly adopted as a cost-effective way to improve model quality across diverse domains. The initial motivation for merging was based on the finding that mode connectivity exists between the loss landscapes of two or more trained neural networks, enabling interpolation of their parameters without incurring an increase in loss (Garipov et al., 2018; Ainsworth et al., 2023; Ito et al., 2024). Simple parameter averaging remains an effective technique; however, more sophisticated strategies based on task vectors have also been proposed to minimize interference in the merged model parameters (Ilharco et al., 2023; Yadav et al., 2023; Yu et al., 2024). Much of the existing literature focuses on the setting in which multiple fine-tunes of a single checkpoint are merged. *Non-local* merging in which the models do not share a common checkpoint is more closely related to expert merging. Sharma et al. (2024) found that re-scaling of model activations was necessary to achieve high-quality non-local merging.

**LLM evaluation.** Evaluating LLMs is challenging; prior work demonstrated that simple metrics such as perplexity can be misleading when used to evaluate compressed LLMs (Jaiswal et al., 2024). MC benchmarks typically measure the log-likelihood of answer tokens to determine a model's response to a question (Gao et al., 2023; Chandak et al., 2025). As such, each response choice is evaluated in a single forward pass, without any tokens being generated by the model. Perplexity and MC accuracy can therefore be viewed as *discriminative* metrics. In contrast, *generative* benchmarks require the model to output a response, more closely corresponding with real-world use-cases of LLMs. Tasks such as code generation, mathematical reasoning with structured outputs, and creative writing are examples of generative benchmarks.

## 3 MOTIVATION

**Setup.** To motivate our proposed expert pruning method, we derive the expected errors of both expert merging and pruning. Consider a SMoE layer with $K$ experts $f_1,...,f_K$, each a function $f_k:\mathbb{R}^d \to \mathbb{R}^d$. Let $\mathcal{T}(x)$ denote the set of indices corresponding to the top-$k$ router scores. The router produces a sparse gating vector $\mathbf{g}(x) \in \mathbb{R}^K_{\geq 0}$ where $g_k(x) > 0$ if $k \in \mathcal{T}(x)$ and $g_k(x) = 0$ otherwise. We assume the active gates are normalized such that $\sum_{k \in \mathcal{T}(x)} g_k(x) = 1$, an operation commonly included in SMoE architectures. The output of the layer is

$$h(x) := \sum_{k \in \mathcal{T}(x)} g_k(x) f_k(x). \tag{1}$$

**Two operations at fixed compression.** To analyse the fundamental difference between compression operations, we focus on the elementary case of reducing two experts, $(f_i, f_j)$, to one by comparing the mean squared reconstruction error, $\mathcal{E} = ||h(x) - \hat{h}(x)||_2^2$ where $\hat{h}(x)$ is output of the layer after compression. *Pruning* removes expert $j$ and re-normalizes the router outputs over the remaining $K-1$ experts. *Merging* replaces $(f_i, f_j)$ with a new expert $\tilde{f}$. Existing one-shot expert merging methods such as HC-SMoE and M-SMoE sum the gates of the original experts $g_i(x) + g_j(x)$. The pruned, $\bar{h}(x)$, and merged, $\tilde{h}(x)$, layer outputs are

$$\bar{h}(x) := \sum_{k \neq j} \frac{g_k(x)}{1 - g_j(x)} f_k(x), \quad (2) \qquad \tilde{h}(x) := \big(g_i(x) + g_j(x)\big)\tilde{f}(x) + \sum_{k \neq i,j} g_k(x) f_k(x). \tag{3}$$

### 3.1 MERGING INDUCES AN INPUT-DEPENDENT TARGET A SINGLE EXPERT CANNOT REALIZE

Define the router's *input-dependent mixing ratio* $r(x) := \frac{g_i(x)}{g_i(x)+g_j(x)} \in [0,1]$ locally on the set where $g_i + g_j > 0$. Substituting $g_i(x)$ and $g_j(x)$ in terms of $r(x)$, the original contribution of the pair $(i,j)$ can be written as

$$g_i(x)f_i(x) + g_j(x)f_j(x) = \big[r(x)(g_i(x)+g_j(x))\big]f_i(x) + \big[(1-r(x))(g_i(x)+g_j(x))\big]f_j(x)$$

$$= \big(g_i(x)+g_j(x)\big)\underbrace{\big(r(x)f_i(x) + (1-r(x))f_j(x)\big)}_{\text{The ideal, input-dependent target expert}}. \tag{4}$$

After merging, the router must apply the summed gate, $g_i(x) + g_j(x)$, to a *constant* convex combination of the constituent experts which is independent of $x$. The core issue is that the merged model is forced to approximate the *dynamic*, input-dependent target expert with a *static* one. The following quantifies this unavoidable approximation error.

**Irreducible error of merging.** Let $\tilde{f}(x) = \alpha f_i(x) + (1-\alpha) f_j(x)$ with a constant $\alpha \in [0,1]$ and define $\Delta_{ij} := f_i(x) - f_j(x)$. This definition of $\tilde{f}$ assumes that the experts are linear functions of $x$ which is generally not the case; however, this simplified model approximates the behaviour of frequency-weighted parameter averaging used by expert merging techniques in practice. $\mathcal{E}_{merge}$ is minimized when $\alpha$ is chosen to be the expected mixing ratio, $\alpha^\star := \mathbb{E}[r(x)]$. Omitting the argument $(x)$ for brevity, this minimal error is

$$\left\| (g_i + g_j)(r f_i + (1-r) f_j) - (g_i + g_j)(\alpha^\star f_i + (1-\alpha^\star) f_j) \right\|^2 = \mathbb{E}_x \Big[ \underbrace{(g_i + g_j)^2}_{\text{router scale}} \cdot \underbrace{(r - \alpha^\star)^2}_{\text{policy variability}} \cdot \underbrace{\|\Delta_{ij}\|^2}_{\text{expert gap}} \Big]. \quad (5)$$

In particular, if the router's policy is not constant ($\mathrm{Var}[r(x)] > 0$) and the experts are not functionally identical ($\|\Delta_{ij}\| > 0$), then every constant-$\alpha$ merge incurs positive error. Let $G_{ij} := \mathbb{E}_x[\|\Delta_{ij}(x)\|_2^2]$. Under a simplifying assumption that the router scale, policy variability, and $G_{ij}$ are weakly correlated across inputs, the error term may be decomposed to:

$$\mathbb{E}_x[(g_i(x) + g_j(x))^2 (r(x) - \alpha^\star)^2 \|\Delta_{ij}(x)\|_2^2] \approx \mathbb{E}_x[(g_i(x) + g_j(x))^2] \cdot \mathrm{Var}[r(x)] \cdot G_{ij} \quad (6)$$

**Consequences.** This is a standard least-squares problem minimized when $\alpha = \mathbb{E}[r]$, and the minimal value is $\mathrm{Var}[r]$. Based on the assumptions noted above, we conclude that merging with summed gates is fundamentally flawed whenever: *(i)* the router has learned an input-dependent policy for mixing two experts ($\mathrm{Var}[r] > 0$); and *(ii)* the experts are themselves distinct ($\|\Delta_{ij}\| > 0$). Any fixed $\alpha$ cannot overcome the irreducible error bound established in Equation (6).

## 3.2 Pruning preserves independent control

Pruning removes one function but importantly does *not* tie the remaining gates. The router still modulates each surviving expert *independently*. In contrast, merging removes a degree of freedom in the policy by replacing individual experts with their mergers. For a direct comparison under no fine-tuning, we consider the error of pruning expert $j$ where $j \in \mathcal{T}(x)$. After pruning, the router promotes previously inactive expert $i$ with the new gate-value of $g_i'(x) \neq 0$, producing the error

$$\mathcal{E}_{prune} = \mathbb{E}_{x|j \in \mathcal{T}(x)} \Big[ \big\| \underbrace{g_j(x) f_j(x) - g_i'(x) f_i(x)}_{\text{substitution error}} - \underbrace{\frac{g_j(x) - g_i'(x)}{1 - g_j(x) + g_i'(x)} \sum_{k \neq i,j} g_k(x) f_k(x)}_{\text{renormalization error}} \big\|_2^2 \Big] \quad (7)$$

Substitution error is the dominant term in the above expression as the renormalization error coefficient simply scales the magnitude of the surviving expert outputs without changing their direction. In contrast, the substitution error includes the output of the promoted expert which may introduce significant error. With top-$k$ routing $g_i' \leq g_j$ and the maximum substitution error occurs when $g_i' \approx g_j$ with a magnitude upper bounded by

$$\|g_j(x) f_j(x) - g_i'(x) f_i(x)\| \leq g_j(x)(\|f_j(x)\| + \|f_i(x)\|). \quad (8)$$

**Synthesis.** While neither method is clearly superior for all distributions, our simplified analysis above isolates specific sources of error. Merging with summed gates couples the experts, incurring error whenever *either* expert is active, unless the experts are functionally identical ($\Delta_{ij} \approx 0$). The router loses the ability to independently modulate the merged experts in an input-dependent manner. Equation (6) establishes that summed gate merging incurs an irreducible error directly proportional to the router's policy variability ($\mathrm{Var}[r(x)]$).

In contrast, pruning only incurs errors when the pruned expert is in the top-$k$ set, $j \in \mathcal{T}(x)$. Unlike Equation (5), Equation (8) *does not* penalize policy variability; the router still controls surviving experts independently. The substitution error from pruning (Eq. 7) is proportional to its gate-value ($g_j$) and is insensitive to policy variability. Highly-granular SMoEs with many experts per layer use highly variable routing policies (high $\mathrm{Var}[r(x)]$) to combine many small contributions (small $g_j(x)$). In this setting, we expect merging with summed gates to be fundamentally disadvantaged.

**Remarks.** (i) The constant-mixture model $\tilde{f}$ is mathematically related to the frequency weighted parameter averaging merge used in practice. (ii) Even if $\tilde{f}$ was dependent on $x$, the router after merging cannot independently modulate the two latent functions, so the original policy is invalidated. (iii) With top-$k$ routers, the specific irreducible error from policy variability ($\mathrm{Var}[r(x)]$) is generated exclusively on the support where *both* experts are selected. Outside that support, this component vanishes, leaving only a static error term that depends on the functional expert gap. (iv) See Appendix A for an extension of the above analysis to hierarchical clustering.

### 3.3 EMPIRICAL EVIDENCE FOR LOSS OF INDEPENDENT CONTROL

**Setup.** We analyse the functional expert output manifolds across four diverse state-of-the-art SMoE architectures by recording mean expert activations from 32 samples of 2048 tokens from the C4 dataset (Raffel et al., 2020).

**Functional subspace collapse.** By projecting expert activations onto their first two principal components, we visualize how pruning and merging affect the learned representations. Figures 1, A5a and A5b demonstrate a striking progression of functional subspace collapse from early to late layers in *high-granularity* architectures such as Qwen3 and ERNIE-4.5. In early layers, the original experts form relatively compact manifolds with moderate spread. After pruning, the surviving experts maintain their positions on the original manifold, preserving its geometric structure with reduced density. In contrast, merging produces a visible contraction toward the manifold's centre. The contrast becomes dramatic in late layers, where experts are more specialized.

The progression from early to late layers validates our theoretical prediction that the irreducible error is proportional to $\mathrm{Var}[r(x)]$. Early layers, which typically learn more generic features, exhibit lower policy variability and thus less dramatic collapse. Late layers, where experts have specialized for distinct computational roles, demonstrate high policy variability, resulting in the severe functional collapse observed when these specialized experts are merged into static averages.

**Functional manifold distortion.** While collapse is less apparent in low-granularity models, the introduction of novel functions due to merging distorts the topology of the original expert manifold to a greater degree than pruning. To quantitatively measure this phenomenon, we measure the 1-Wasserstein distance (Earth Mover's distance) between the original and compressed expert output manifolds, see Appendix B.2 for details. As depicted in Figure 2, the merged outputs consistently exhibit a higher transport cost from the original manifold.

**Manifold geometry preservation.** Across all models and layers, we observe that pruning preserves the topology of the functional manifold while merging fundamentally alters it. The preservation of manifold geometry under pruning reflects the mathematical structure of the operation: the pruned expert class is a co-ordinate subspace of the original, with the router maintaining independent control over each surviving expert.

In contrast, the subspace collapse observed in merged highly-granular SMoEs visualizes the loss of independent control. When gates $g_i$ and $g_j$ are tied by their sum $(g_i + g_j)$, the router can no longer independently modulate the two underlying functions, forcing the model to approximate the dynamic mixture $r(x)f_i(x) + (1-r(x))f_j(x)$ with a static merged expert $\tilde{f}$.

With low-granularity SMoEs, such as Llama-4-Scout and Mixtral, functional subspace collapse due to expert merging is less apparent, see Figures A5c to A5f. With few experts per layer and active experts per token, these architectures have less variable routing and higher gate-values, which better preserves the variance of the original manifold. However, the introduction of novel functions by merging introduces greater manifold distortion than the substitution error associated with pruning. These observations reveal that the core issue is not the reduction in the number of experts *per se*, but rather the qualitative change in the router's control structure and the introduction of novel functionality. See Appendix B for additional discussion.

## 4 ROUTER-WEIGHTED EXPERT ACTIVATION PRUNING (REAP)

The motivation in Section 3 demonstrates that the functional output space of a SMoE layer is defined by the *coordinated behaviour* of the router and experts. As established in Equation (8), the magnitude of the substitution error incurred by promoting expert $i$ in lieu of pruned expert $j$ is upper bounded by $g_j(x)(\|f_j(x)\| + \|f_i(x)\|)$. Naive frequency-based pruning considers neither the coordination between

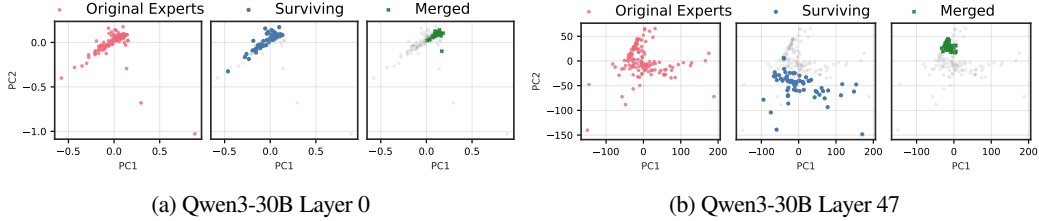

(a) Qwen3-30B Layer 0            (b) Qwen3-30B Layer 47

Figure 1: (a) **Functional subspace (PCA) for early SMoE layers in Qwen3-30B**. Pruning (blue) preserves the manifold geometry; merging (green) collapses it toward the centre. (b) **Functional subspace (PCA) for late MoE layers.** The contraction under merging is dramatically more pronounced, with up to $100\times$ reduction in spread for models with many experts. See Figure A5 for results from other models.

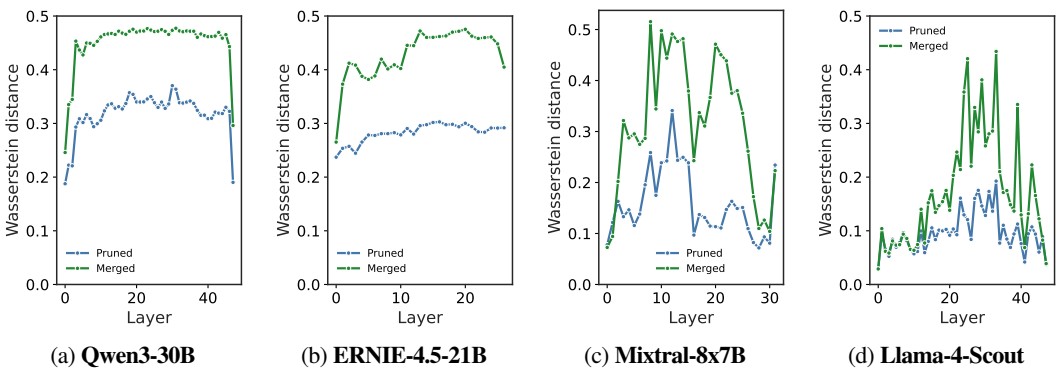

(a) **Qwen3-30B**      (b) **ERNIE-4.5-21B**      (c) **Mixtral-8x7B**      (d) **Llama-4-Scout**

Figure 2: 1-**Wasserstein distance** between the compressed and original expert output manifolds measured in normalized angular distance. Expert merging introduces novel functions which distort the manifold.

router and expert ($g_j(x)$) nor the functional properties of the pruned expert ($\|f_j(x)\|$), effectively assuming that all active experts contribute equally to the output. By ignoring these terms, frequency-based methods fail to minimize the error bound derived above.

Since the identity of the promoted expert $i$ (and thus $\|f_i(x)\|$) varies across tokens, directly minimizing the pruned expert's impact $g_j(x)\|f_j(x)\|$ is an effective heuristic to minimize the total error. This strategy targets the known components of the error bound ($g_j\|f_j\|$) while simultaneously shrinking the scaling coefficient ($g_j$) of the unknown component ($\|f_i\|$). Intuitively, this identifies experts which contribute minimally to the layer output, yielding the minimal difference between the original and pruned layer outputs in expectation. To select which experts to prune, we propose a novel saliency criterion, REAP. Specifically, the saliency score, $S_j$, is defined as the average of the expert's weighted magnitude over tokens for which it is active:

$$S_j = \frac{1}{|\mathcal{X}_j|} \sum_{x \in \mathcal{X}_j} g_j(x) \cdot \left\| f_j(x) \right\|_2, \tag{9}$$

where $\mathcal{X}_j$ is the set of tokens where expert $j$ is active (i.e., $\mathcal{X}_j = \{x \mid j \in \mathcal{T}(x)\}$). Crucially, calculating this average conditionally over $\mathcal{X}_j$ rather than globally decouples the expert's functional impact from its frequency of activation. A global average may be dominated by usage frequency and risks pruning *specialist* experts which are rarely activated but contribute significantly to the layer output when selected. By pruning experts with the lowest $S_j$, REAP targets those that provide a weak functional contribution even when specifically requested by the router, thereby minimizing the substitution error bound for every active token.

## 5 EXPERIMENTS

**Setup.** We implement REAP and other expert compression baselines in PyTorch (Ansel et al., 2024). We collect router logits and expert activation data to calibrate the compression algorithms using a variety of general pre-training and domain-specific Supervised Fine-Tuning (SFT) datasets. For calibration, 1,024

Table 1: Comparison of SMoE models included in our study.

| Model | Routed Experts | Shared Experts | Top-K | Sparsity | Parameters (1e9) | Active Params. (1e9) | First layer dense | Expert Granularity |
|---|---|---|---|---|---|---|---|---|
| ERNIE-4.5-21B-A3B-PT | 64 | 2 | 6 | 87.88% | 21.9 | 3 | Yes | High |
| Qwen3-30B-A3B | 128 | 0 | 8 | 93.75% | 30.5 | 3 | No | High |
| Mixtral-8x7B-Instruct-v0.1 | 8 | 0 | 2 | 75.00% | 46.7 | 13 | No | Low |
| GLM-4.5-Air | 128 | 1 | 8 | 93.02% | 106.9 | 12 | Yes | High |
| Llama-4-Scout-17B-16E-Instruct | 16 | 1 | 1 | 88.24% | 107.8 | 17 | No | Low |
| Qwen3-Coder-480B-A35B-Instruct-FP8 | 160 | 0 | 8 | 95.00% | 480.2 | 35 | No | High |
| Kimi-K2-Instruct-W4A16 (RedHatAI, 2025) | 384 | 1 | 8 | 97.66% | 1026.4 | 32 | Yes | High |

samples are randomly selected and packed to 2,048 sequence length for models with $\leq$ 110B parameters. For models with $\geq$ 110B parameters, we select 12,228 samples with a maximum sequence length of 16,384 tokens without truncation or packing.

We compress models by pruning or merging 25% or 50% of experts in each layer, except for M-SMoE which determines the number of clusters per layer based on global expert usage frequency. When evaluating models with $\leq$ 50B parameters on coding and MC, we calibrate and compress the models using three different seeds and report the mean. Larger models, creative writing, and mathematical reasoning evaluations are reported using a single seed, except where explicitly noted otherwise. All models are evaluated in the one-shot setting, with no additional fine-tuning after compression.

**Models and data.** We evaluate the expert compression algorithms on a diverse set of six SMoE architectures covering model sizes from 21B to 1T with varying degrees of sparsity and expert granularity, see Table 1 for details. For MC question answering and code generation benchmarks, we use C4 (Raffel et al., 2020; Allen Institute for AI, 2024) and evol-codealpaca (Chaudhary, 2023; Luo et al., 2024; Tam, 2023) datasets to assess both general and domain-specific calibration. Models with $\geq$ 110B parameters are additionally calibrated with data from xlam-function-calling (Liu et al., 2024c; Salesforce, 2025) and SWE-smith-trajectories (Yang et al., 2025c;b) datasets. For creative writing and math benchmarks we employ WritingPrompts curated (Pritsker, 2024) and tulu-3-sft-personas-math (Lambert et al., 2025; Allen Institute for AI, 2025), respectively. The default chat template is applied to all SFT datasets and `</think>` tags are explicitly closed to disable reasoning in hybrid reasoning models.

**Evaluation.** Compressed SMoE models are evaluated on a suite of benchmarks including MC question answering, code generation, mathematical reasoning, creative writing, and tool calling. See Appendix C for details. We implement HC-SMoE and M-SMoE as expert merging baselines. Average linkage criterion is used for HC-SMoE. M-SMoE does not include low-rank compression from the complete MC-SMoE method. Pruning baselines consist of frequency-based pruning and EAN and experts with the lowest saliency scores according to each method's criterion are pruned. See Appendix D for formal definitions.

## 5.1 RESULTS

In Table 2 and Figure 3 code generation, creative writing, math reasoning, and MC results are presented for Qwen3-30B and GLM-4.5-Air after calibration with domain-specific datasets. Table 3 contains results for large-scale SMoE pruned models on code generation, tool calling, and MC benchmarks. See Table A6 and Table A7 for detailed MC and code generation results, respectively. Figure A6 depicts coding generation and MC accuracy versus model parameters. See Appendix F for additional results.

**Zero-shot MC question answering.** Both merging and pruning are capable of producing accurate compressed SMoE models for MC question answering. HC-SMoE and REAP have a mean decrease in accuracy of approximately 4% and 13% for compression ratios of 25% and 50%, respectively, excluding large-scale SMoEs. REAP achieves first or second rank among all methods, models and compression ratios, suggesting strong consistency regardless of specific model architecture. When calibrated on C4, we find slightly improved accuracies for all compression methods with similar rankings as noted above, see Table A8.

**Generative benchmarks.** Compared to MC, generative benchmarks are more representative of real-world use cases of LLMs. In this setting, pruning emerges as the clearly superior compression method on the generative task benchmarks. Excluding large-scale SMoEs, REAP achieves a mean decrease in accuracy of 1.9% and 6.9% at 25% and 50% compression ratios, respectively, on coding. In comparison, both HC-SMoE and

Table 2: MC and generative benchmark results for Qwen3-30B and GLM-4.5-Air.

| Model | Compression | Technique | Method | Coding Eval+ | LiveCode | Code Avg | Creative Writing WildBench | GSM8K | Math MATH-500 | Math Avg | MC MC Avg |
|---|---|---|---|---|---|---|---|---|---|---|---|
| Qwen3-30B-A3B | Baseline | | | 0.814 | 0.302 | 0.558 | 0.811 | 0.903 | 0.872 | 0.887 | 0.721 |
| | 25% | Merging | M-SMoE | 0.781 | 0.293 | 0.537 | 0.805 | 0.901 | 0.872 | 0.886 | 0.558 |
| | | | HC-SMoE | 0.752 | 0.258 | 0.505 | 0.497 | 0.864 | 0.834 | 0.849 | **0.674** |
| | | Pruning | Frequency | 0.805 | 0.302 | 0.553 | 0.807 | 0.910 | 0.865 | 0.888 | 0.600 |
| | | | EAN | 0.797 | 0.311 | **0.554** | **0.811** | 0.904 | 0.879 | **0.892** | 0.603 |
| | | | REAP | 0.797 | 0.304 | 0.551 | 0.804 | 0.896 | 0.881 | 0.888 | 0.665 |
| | 50% | Merging | M-SMoE | 0.590 | 0.205 | 0.397 | **0.725** | 0.824 | 0.838 | 0.831 | 0.451 |
| | | | HC-SMoE | 0.543 | 0.185 | 0.364 | 0.008 | 0.760 | 0.696 | 0.728 | **0.542** |
| | | Pruning | Frequency | 0.668 | 0.236 | 0.452 | 0.677 | 0.871 | 0.860 | **0.865** | 0.483 |
| | | | EAN | 0.753 | 0.306 | 0.530 | 0.702 | 0.874 | 0.855 | 0.864 | 0.493 |
| | | | REAP | 0.780 | 0.302 | **0.541** | 0.718 | 0.877 | 0.838 | 0.857 | 0.503 |
| GLM-4.5-Air | Baseline | | | 0.786 | 0.374 | 0.580 | 0.839 | 0.846 | 0.918 | 0.882 | 0.747 |
| | 25% | Merging | M-SMoE | 0.726 | 0.330 | 0.528 | 0.781 | 0.848 | 0.880 | 0.864 | 0.596 |
| | | | HC-SMoE | 0.737 | 0.363 | 0.550 | 0.788 | 0.842 | 0.908 | 0.875 | **0.704** |
| | | Pruning | Frequency | 0.759 | 0.341 | 0.550 | 0.793 | 0.832 | 0.908 | 0.870 | 0.648 |
| | | | EAN | 0.768 | 0.374 | 0.571 | 0.824 | 0.839 | 0.908 | 0.874 | 0.637 |
| | | | REAP | 0.759 | 0.412 | **0.585** | **0.831** | 0.839 | 0.920 | **0.879** | 0.674 |
| | 50% | Merging | M-SMoE | 0.468 | 0.099 | 0.284 | 0.391 | 0.465 | 0.466 | 0.465 | 0.444 |
| | | | HC-SMoE | 0.618 | 0.220 | 0.419 | 0.593 | 0.667 | 0.732 | 0.700 | **0.564** |
| | | Pruning | Frequency | 0.511 | 0.104 | 0.308 | 0.604 | 0.615 | 0.612 | 0.613 | 0.521 |
| | | | EAN | 0.721 | 0.253 | 0.487 | 0.702 | 0.781 | 0.838 | 0.809 | 0.511 |
| | | | REAP | 0.679 | 0.352 | **0.515** | **0.754** | 0.815 | 0.900 | **0.857** | 0.554 |

Table 3: Large-scale pruned SMoEs on agentic, non-agentic coding, tool-use tasks, and MC benchmarks.

| Model | Compression | Method | Non-Agentic Coding Eval+ | LiveCode | Code Avg | Agentic Coding SWE-Bench-Verified | Tool-Use (BFCLv3) Non-Live | Live | Multi-Turn | Overall | MC MC Avg |
|---|---|---|---|---|---|---|---|---|---|---|---|
| Qwen3-Coder-480B-A35B-Instruct-FP8 | Baseline | | 0.841 | 0.431 | 0.636 | 0.540 | 0.866 | 0.825 | 0.380 | 0.690 | 0.750 |
| | 25% | Frequency | 0.737 | 0.296 | 0.516 | 0.378 | 0.844 | 0.763 | 0.355 | 0.654 | 0.606 |
| | | EAN | 0.827 | 0.419 | 0.623 | 0.534 | 0.831 | 0.813 | 0.384 | 0.676 | 0.702 |
| | | REAP | 0.831 | 0.416 | **0.624** | **0.540** | 0.878 | 0.823 | 0.392 | **0.698** | **0.748** |
| | 50% | Frequency | 0.007 | 0.012 | 0.010 | 0.000 | 0.200 | 0.392 | 0.000 | 0.197 | 0.506 |
| | | EAN | 0.777 | 0.382 | 0.580 | **0.536** | 0.822 | 0.774 | 0.383 | 0.659 | 0.591 |
| | | REAP | 0.822 | 0.415 | **0.619** | 0.522 | 0.849 | 0.801 | 0.371 | **0.674** | **0.692** |
| Kimi-K2-Instruct-W4A16 | Baseline | | 0.828 | 0.434 | 0.631 | 0.554 | 0.840 | 0.802 | 0.355 | 0.666 | 0.780 |
| | 25% | Frequency | 0.486 | 0.082 | 0.284 | 0.000 | 0.644 | 0.603 | 0.045 | 0.431 | 0.604 |
| | | EAN | 0.779 | 0.379 | 0.579 | 0.562 | 0.819 | 0.802 | 0.335 | **0.652** | 0.703 |
| | | REAP | 0.840 | 0.440 | **0.640** | **0.580** | 0.842 | 0.801 | 0.263 | 0.635 | **0.773** |
| | 50% | Frequency | 0.112 | 0.000 | 0.056 | 0.000 | 0.255 | 0.397 | 0.003 | 0.218 | 0.439 |
| | | EAN | 0.722 | 0.253 | 0.487 | **0.576** | 0.778 | 0.767 | 0.173 | **0.573** | 0.587 |
| | | REAP | 0.819 | 0.429 | **0.624** | **0.576** | 0.785 | 0.743 | 0.164 | 0.564 | **0.643** |

M-SMoE produce mean decreases in accuracy >5% at 25% compression and >20% at 50% compression. Notably, REAP maintains significantly higher accuracy at 50% compression than other pruning methods. M-SMoE achieves significantly better code generation accuracy on low-granularity SMoE architectures.

On creative writing, REAP and EAN are near-lossless at 25% compression with REAP offering improved quality at 50% compression. Merging methods are less consistent across various model architectures and compression ratios. For example, M-SMoE is the best method for Qwen3-30B at 50% compression, but the worst on GLM-4.5-Air. REAP attains the best mathematical reasoning results with a remarkable mean decrease in accuracy of just 0.1% at 25% compression. In comparison, HC-SMoE and M-SMoE offer high accuracy at 25% compression but are significantly less accurate than pruning at 50% compression.

**Expert pruning at scale.** To assess whether pruning remains viable at scale, we prune Qwen3-Coder-480B and Kimi-K2-Instruct. On MC questions, REAP outperforms other pruning methods. On non-agentic coding tasks, REAP achieves near-lossless accuracy with a 0.16% and 1.2% mean decrease in accuracy compared to baseline at 25% and 50%, respectively, outperforming EAN and frequency-based pruning, particularly at 50% compression. On the challenging SWE-Bench task, both REAP and EAN maintain high accuracy at 25% and 50% compression, with some scores slightly exceeding the baseline. On tool use, EAN and REAP are comparable, with REAP slightly outperforming at 50% compression with a mean decrease in accuracy of 5.9% versus 6.2% for EAN. Frequency-based pruning suffers from a sharp degradation in quality at 50% compression, highlighting the importance of pruning saliency criteria which consider expert activations. Scaling the pruning methods is relatively trivial. Unlike HC-SMoE, calibration for pruning does not require recording activations from every expert for every token, facilitating efficient calibration. Further, pruning can be easily applied to quantized models without any additional steps required to reconcile block scales or re-quantize following compression. See Appendix E for further discussion.

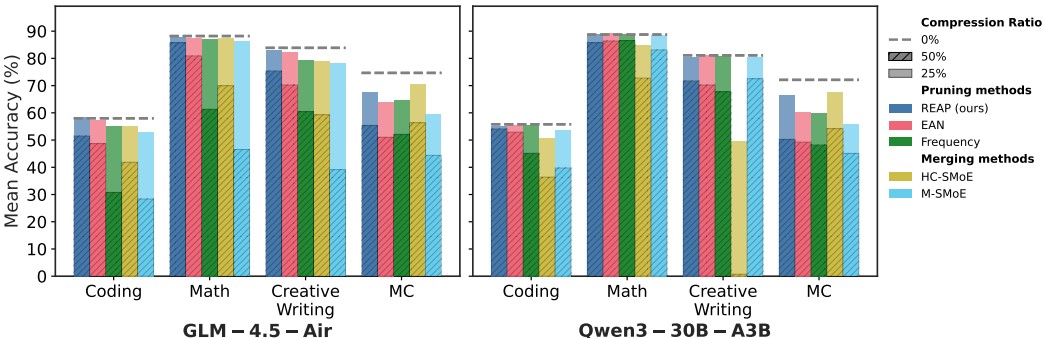

Figure 3: **GLM-4.5-Air and Qwen3-30B accuracy vs. task type.** REAP offers significant improvements compared to other methods at 50% compression. Note the significant performance drop for merging methods on generative tasks (Coding, Math, Creative Writing) compared to their relative strength on MC.

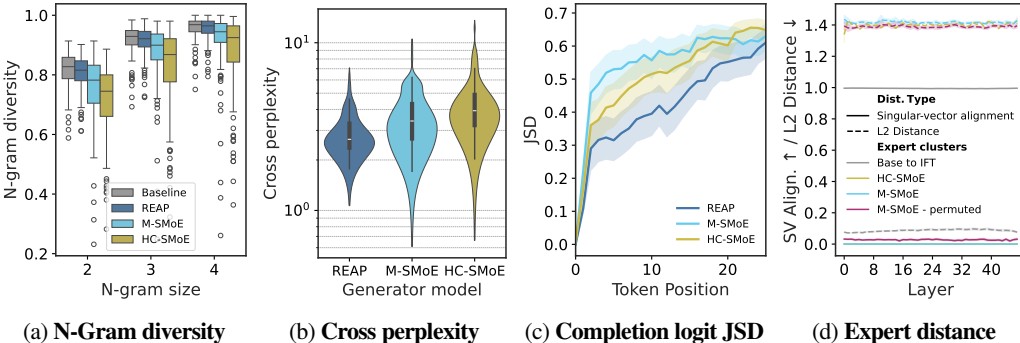

| (a) **N-Gram diversity** | (b) **Cross perplexity** | (c) **Completion logit JSD** | (d) **Expert distance** |

Figure 4: (a) & (b) **N-Gram diversity** and **cross-perplexity** of compressed Qwen3-30B-A3B models at 50% compression, respectively. (c) **Jensen-Shannon Divergence (JSD) of compressed and baseline model logits vs. completion token position** for Qwen3-30B-A3B at 50% compression. Initially, all compressed models share close alignment with the baseline model. However, as the completion token position increases the merged models diverge from the baseline more rapidly than the REAP pruned model. (d) The mean relative **L2-distance** and **singular-vector alignment** between Qwen3-30B expert weights at 50% compression. Expert merging is more challenging than model merging due to large distances between experts in weight space and low singular-vector alignment.

**Quantifying merged SMoE generation quality.** While merged expert SMoEs offer reasonable quality for discriminative tasks such as MC question and answering, they fail to remain competitive on generative tasks. To help explain the performance gap of merged models between discriminative and generative tasks, we perform an analysis of the compressed model outputs and compare with REAP pruned models. We prompt 50% compressed Qwen3-30B models with 100 questions randomly sampled from the evol-codealpaca dataset and record their outputs. In Figure 4a, we measure the N-gram diversity and find that the merged models have significantly lower diversity across all N-gram sizes measured. In contrast, the REAP pruned model remains similar to the base model, albeit slightly less diverse. In Figure 4b, we measure the perplexity of the text generated by the compressed models with the original baseline model. The text generated by the merged models has both a higher mean and higher variance than the pruned model generations, suggesting that the REAP pruned model outputs are more closely aligned to the original model. The alignment between the baseline and REAP pruned SMoEs is further supported by Figure 4c, which plots the JSD of the compressed and baseline logits vs. output token position. The merged model logits diverge from the baseline more rapidly than the pruned model.

**The challenges of expert merging.** Model merging has been widely adopted to facilitate LLM fine-tuning. Why does expert merging miss the mark? In addition to the loss of the router's input-dependent modulation of experts explored in Section 3, we argue that the non-local nature of expert merging and high cardinality of expert clusters pose significant unresolved challenges. In Figure 4d, we plot the mean relative L2-distance between experts clustered by HC-SMoE or M-SMoE and compare with the distance between expert weights from the pretrained to Instruct Fine-Tuned (IFT) checkpoints. We find that the distance between

clustered experts within the same layer greatly exceeds that of experts in the IFT checkpoint after fine-tuning. Ito et al. (2024) found that weight matching permutations improved alignment of parameters' singular vectors. Following their approach, we decompose expert weights with Singular Value Decomposition (SVD) and plot the singular-vector alignment in Figure 4d. Even after applying weight matching permutations, the M-SMoE expert clusters remain far apart both in weight space and singular-vector alignment. The relatively poorly aligned experts highlight the considerable challenge of coherently merging their parameters.

When merging works well, it's more closely related to pruning than one might expect. In Figure A7a, we depict the frequency of singleton clusters — clusters containing a single expert — for both HC-SMoE and M-SMoE. A singleton cluster is directly analogous to an expert that remains after pruning. We find that HC-SMoE in particular has a high prevalence of singleton clusters, leaving important experts unadulterated and compressing the rest into a few *mega*-clusters containing tens of experts. This is particularly true of the high granularity models which contain more experts per layer. We hypothesize that the cardinality of these mega-clusters poses a challenge for existing merging algorithms and test this intuition in Figure A7b. Unfortunately, even modest restrictions of the maximum cluster size to 32 — half the number of experts to compress — results in large decreases in model quality on coding tasks.

**The importance of domain-specific calibration.** In Figure A8, we plot the code generation accuracy of the various compression methods and models when calibrated on either C4 or evol-codealpaca. The difference is stark, C4 calibration results in a collapse in accuracy, with several compressed model instances failing to produce coherent outputs, resulting in 0% accuracy. In Figure A9, we compare the accuracy of compressed Qwen3-30B models calibrated with either domain-specific data or the combined calibration data across all generative tasks. While domain-specific calibration results in higher accuracies, REAP best preserves accuracy compared to other compression methods in the combined data calibration setting.

## 6 DISCUSSION

Similar to prior work, we find that expert merging performs reasonably well on MC benchmarks. This may be because MC tasks only require a discriminative function that can be approximated by an *average* expert. In contrast, merging fails to maintain model quality on generative tasks, particularly at 50% compression and high-granularity architectures. Generative tasks require auto-regressive generation, a capability that is impaired when the router's fine-grained control is removed or novel expert functions are introduced. Compared to expert pruning, merging is less consistent, exhibiting higher variance across models and compression ratios. The outputs of expert merged models are more repetitive and less closely aligned with the base model compared with pruned model's outputs. Taken together, these observations are direct evidence of functional manifold distortion of the SMoE layers discussed in Section 3.3.

Overall, expert pruned models offer consistently higher accuracy than merged models on generative tasks. REAP is a robust pruning criterion that generalizes across a wide array of SMoE architectures, compression ratios, and generative tasks. By taking into consideration both the router gate-values and expert activation norms, REAP minimizes the reconstruction error bound by pruning experts which contribute the least to each layers output. REAP is scalable, achieving near-lossless compression on coding tasks with Qwen3-Coder-480B and Kimi-K2. The successes of REAP highlight the crucial importance of preserving coordination between the router and experts. Compression methods which impair the router's ability to independently modulate expert outputs or distort the original functional manifold are less likely to succeed.

Finally, this work highlights the importance of comprehensive downstream evaluations and the significant challenges involved with evaluating LLMs. Discriminative metrics such as perplexity and log-likelihood based MC benchmarks are not necessarily good proxies for generative model quality.

## 7 CONCLUSION

Our analysis of current SMoE expert merging techniques introduces irreducible error due to the loss of the router's independent control over experts. In contrast, expert pruning produces a coordinate subspace of the original layer which maintains the topology of the functional manifold. We introduce REAP, a novel expert pruning method which prunes experts that contribute the least to the layer's output, thereby minimizing the reconstruction error bound. Empirically, we demonstrate that REAP retains remarkably high accuracy on a wide array of generative tasks across a diverse set of model architectures. We hope that this work inspires further compression techniques for SMoEs and facilitates the deployment of accurate, domain-specific models in resource constrained settings.

## ACKNOWLEDGMENTS

We would like to acknowledge the helpful feedback of Valavan Manohararajah, Mohammed Adnan, and Rohan Jain. This work was supported in part by RBC Borealis through the RBC Borealis AI Global Fellowship Award. ML and YI gratefully acknowledge the support of Alberta Innovates (ALLRP 577350-22, ALLRP 600038-24), the Natural Sciences and Engineering Research Council of Canada (NSERC) (RGPIN-2022-03120, DGECR-2022-00358), Defence Research and Development Canada (DGDND-2022-03120), and NSERC/Agence Nationale de la Recherche (ANR) (ALLRP 602719-24). This project was supported thanks to funding from IVADO and the Canada First Research Excellence Fund. This research was enabled in part by support provided by the Digital Research Alliance of Canada (alliancecan.ca). YI is supported by a Schulich Research Chair.

## ETHICS STATEMENT

This research focused on the algorithmic compression of SMoE models and does not involve the use of human subjects, personally identifiable information, or sensitive data. The datasets used for calibration and evaluation (e.g., C4, evol-codealpaca) are publicly available. Our aim is to enable the use of large-scale SMoE models in resource constrained settings. However, we acknowledge that compression techniques such as REAP could potentially facilitate deployment of models for malicious purposes. Further, our compression methods are applied to pre-trained models and any biases related to fairness, discrimination, or representation inherent in the original models may be present in their compressed versions. We make no attempt in this work to mitigate these potential biases. The primary contribution of this paper is technical, and we do not foresee any new, direct ethical concerns arising from our proposed methodology beyond those already associated with the deployment of large language models.

## REPRODUCIBILITY STATEMENT

We are committed to ensuring the reproducibility of our research. We have open-sourced our code and released select compressed model checkpoints to facilitate further research on compressed SMoEs. REAP is formally described in Section 4. The baseline methods we compare against, including frequency-based pruning, EAN, M-SMoE, and HC-SMoE, are formally defined in Appendix D. Section 5 provides a detailed description of our experimental setup, including the specific models used, the calibration and evaluation datasets, and the implementation details for all compression experiments. Further evaluation details are provided in Appendix C.

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

## A  EXTENSION TO HIERARCHICAL CLUSTERING

While Equation (5) analyses pairwise merging, practical implementations often employ hierarchical clustering to form groups of experts. Consider a cluster $C = \{f_{i_1}, ..., f_{i_k}\}$ of $k$ experts merged into a single representative $\tilde{f}_C$. The original contribution of this cluster can be decomposed as:

$$\sum_{j \in C} g_{i_j}(x) f_{i_j}(x) = \left( \sum_{j \in C} g_{i_j}(x) \right) \cdot \underbrace{\sum_{j \in C} w_j(x) f_{i_j}(x)}_{\text{Dynamic, input-dependent mixture}} \tag{10}$$

where $w_j(x) = \frac{g_{i_j}(x)}{\sum_{l \in C} g_{i_l}(x)}$ are the within-cluster mixing ratios that sum to 1.

After hierarchical merging, the router must apply the *summed gate* $\sum_{j \in C} g_{i_j}$ to a *single, static* cluster representative $\tilde{f}_C$, typically computed as a weighted average of the cluster members based on calibration data. This induces an irreducible error.

**Hierarchical clustering error.**  For a cluster $C$ merged into $\tilde{f}_C = \sum_{j \in C} \alpha_j f_{i_j}$ with fixed weights $\alpha_j \geq 0$, $\sum_j \alpha_j = 1$, the minimal $L^2$ error is:

$$\min_{\{\alpha_j\}} \left\| \sum_{j \in C} g_{i_j} f_{i_j} - \left( \sum_{j \in C} g_{i_j} \right) \tilde{f}_C \right\|^2 = \mathbb{E}\left[ \left( \sum_{j \in C} g_{i_j} \right)^2 \right] \cdot \text{Var}_x\left[ \sum_{j \in C} w_j(x) f_{i_j}(x) \right] \tag{11}$$

The error grows with both the cluster's total gate-value and the variance of the dynamic mixture that the cluster must approximate with a static representative.

**Implications for cluster formation.**  The hierarchical error bound reveals a fundamental tension:

- **Large clusters** ($|C|$ large) aggregate more gate-value $\sum_{j \in C} g_{i_j}$, amplifying any approximation error
- **Diverse clusters** (high $\|\Delta_{ij}\|$ for $i, j \in C$) increase the variance term, as the static representative must approximate a wider range of functions
- **Imbalanced clustering** (many singletons, few mega-clusters) combines the worst aspects: mega-clusters suffer severe collapse while singletons provide minimal compression

Distance metrics like Euclidean distance that consider magnitude can exacerbate these issues by creating clusters based on norm similarity rather than functional role, potentially grouping experts with different specializations but similar scales. The resulting mega-clusters force the router to apply a single control signal to what were previously dozens of independently modulated experts, explaining the catastrophic functional collapse observed empirically in late layers where $\text{Var}[w_j(x)]$ is highest.

## B  ADDITIONAL EMPIRICAL EVIDENCE FOR LOSS OF INDEPENDENT CONTROL

### B.1  FUNCTIONAL SUBSPACE PCA ANALYSIS

**Qualitative evidence of functional subspace collapse.**  In Figure 1a, Qwen3's layer 0 exemplifies the contraction of the functional output space by merging in early layers. The original 128 experts span from $-0.4$ to $1.0$ along PC1, pruning maintains this full range with 64 experts, while merging contracts the distribution to approximately $[-0.2, 0.3]$, a 5-fold reduction. This contraction is dramatic in late layers, where experts are more specialized, as can be seen in Figure 1b. Figures A5a and A5b exhibit similar contractions of the expert output manifold under merging, whereas pruning often preserves outlier experts and the span of the original expert output manifold.

In Table A4, we tabulate the total cumulative variance explained by PC1+PC2 for the PCA projections in Figures 1 and A5. For low-granularity SMoEs such as Llama-4-Scout and Mixtral, PC1 and PC2 capture most of the variance in the activations. Even in high-granularity SMoEs such as Qwen3 and ERNIE, a large portion of the total variance is captured by PC1 and PC2. The merged variance explained is consistently higher than the baseline, suggesting that the merged outputs have lost some of their high-dimensional

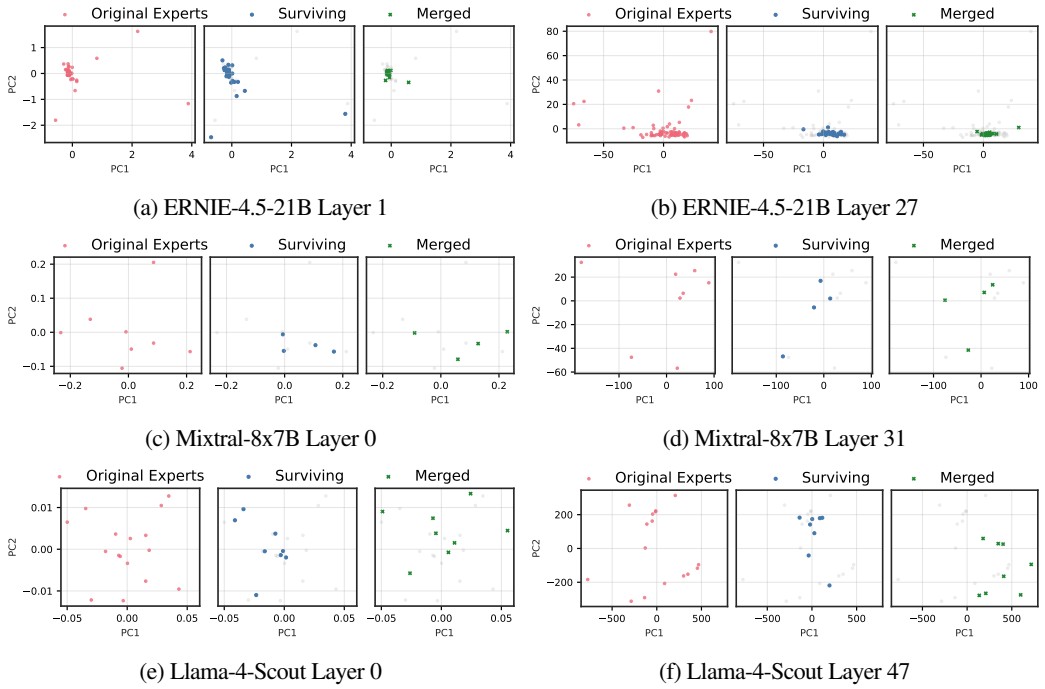

Figure A5: (a,c,e) **Functional subspace (PCA) for early SMoE layers**. Pruning (blue) preserves the manifold geometry; merging (green) collapses it toward the centre. (b,d,f) **Functional subspace (PCA) for late MoE layers.**

complexity. In contrast, the pruned variance explained is consistently lower than the baseline, suggesting that pruning preserves outlier experts and the high-dimensional complexity of the baseline model.

**The role of expert granularity.** Both Qwen3-30B-A3B and ERNIE-4.5-21B are *highly-granular* SMoEs containing 128 and 64 routed experts per layer, respectively, and 8 and 6 routed experts per token, respectively. Functional subspace collapse due to expert merging is more pronounced in these models than in *low-granularity* models such as Mixtral-8x7B and Llama-4-Scout. With fewer experts and a lower amount of experts per token, low-granularity SMoEs appear to better preserve the variance of their expert output manifolds under expert merging. For example, in Figure A5c, the merged manifold spans along PC1 from approximately $[-0.1, 0.2]$ whereas the pruned manifold spans from approximately $[0.0, 0.2]$ along PC1. Similarly, as depicted in Figure A5e, the pruned and merged manifolds span PC1 along $[-0.04, 0.0]$ and $[-0.05, 0.05]$, respectively.

However, the merged manifold is distorted by the introduction of novel expert functions. For example, in Figure A5e, expert merging introduces novel functions which occupy approximately $[0.05, 0.005]$ and $[-0.025, 0.005]$ which are significantly different than any of the original experts. This is best exemplified by Figure 2, which plots the Wasserstein distance between the original and compressed expert output manifolds in terms of normalized angular distance. Compared to the pruned models, the higher distances between the merged and original expert output manifolds suggest a lower degree of similarity. The distorted manifold of the merged expert outputs represents a loss of fidelity with the original manifold which cannot be restored in the one-shot compression setting.

## B.2 WASSERSTEIN DISTANCE

To quantify the distortion of the original expert output manifold, we measure the 1-Wasserstein distance (Earth Mover's distance) between the original and compressed expert output manifolds, see Figure 2.

The distance is calculated between the discrete empirical distributions of the original expert outputs, $\mathcal{F} = \{f_1, ..., f_K\}$, and compressed expert outputs, $\hat{\mathcal{F}} = \{\hat{f}_1, ..., \hat{f}_{K/2}\}$, projected onto the unit hypersphere

Table A4: **Cumulative variance explained** by PC1 and PC2 across compression methods. Compared to pruning, merging results in a consistently higher explained variance suggesting that the merged models have lost some of their high-dimensional complexity.

| Model | Layer | Baseline | Merged | Pruned |
|---|---|---|---|---|
| Qwen3-30B-A3B | 0 | 0.2343 | 0.2700 | 0.1845 |
| Qwen3-30B-A3B | 47 | 0.7195 | 0.7437 | 0.6860 |
| ERNIE-4.5-21B | 0 | 0.3836 | 0.2851 | 0.2733 |
| ERNIE-4.5-21B | 26 | 0.2563 | 0.4599 | 0.0785 |
| Llama-4-Scout | 0 | 0.9032 | 0.9343 | 0.8480 |
| Llama-4-Scout | 47 | 0.9473 | 0.9546 | 0.8754 |
| Mixtral-8x7B | 0 | 0.6486 | 0.8479 | 0.4016 |
| Mixtral-8x7B | 31 | 0.8580 | 0.8140 | 0.7027 |

$$W_1(\mathcal{F},\hat{\mathcal{F}}) = \inf_{\gamma \in \Gamma(\mu,\nu)} \sum_{i=1}^{K} \sum_{j=1}^{K/2} \gamma_{ij} \frac{1}{\pi} \arccos\left( \frac{f_i \cdot \hat{f}_j}{\|f_i\| \|\hat{f}_j\|} \right) \tag{12}$$

where $\mu$ and $\nu$ are uniform probability measures over the indices of $\mathcal{F}$ and $\hat{\mathcal{F}}$ respectively, $\Gamma(\mu,\nu)$ is the set of all transport plans (joint distributions) with marginals $\mu$ and $\nu$, and the cost function is defined as the normalized angular distance. This metric quantifies the minimum "work" required to transport the probability mass from the compressed functional manifold to cover the original manifold, thereby penalizing both the contraction of variance (subspace collapse) and the introduction of functionally distinct artifacts (distortion).

## C  EVALUATION DETAILS

**Multiple choice (MC) evaluation.** Following Chen et al. (2025), our MC benchmarks include: AI2 Reasoning Challenge (ARC-c & ARC-e) (Clark et al., 2018), BoolQ (Clark et al., 2019), HellaSwag (Zellers et al., 2019), MMLU (Hendrycks et al., 2021a), OpenBookQA (OBQA) (Mihaylov et al., 2018), Recognizing Textual Entailment Challenge (RTE) (Bentivogli et al., 2009), and WinoGrande (WinoG.) (Sakaguchi et al., 2021). We evaluate the models in the zero-shot setting using the standard log-likelihood approach with lm-eval-harness (Gao et al., 2023). We report byte-length normalized accuracies for ARC-c, ARC-e, HellaSwag, and OBQA[1].

**Coding evaluation.** For code generation, all models are evaluated on EvalPlus (Liu et al., 2023) and 182 LiveCodeBench (Jain et al., 2025) questions collected between January and April 2025. We extend the original source code for these benchmarks to evaluate our models. We additionally evaluate Kimi-K2-Instruct-W4A16 and Qwen3-Coder-480B on the agentic coding benchmark SWE-Bench (Jimenez et al., 2024) and tool-calling benchmark BFCLv3 (Patil et al., 2025). For BFCLv3, we use the original Gorilla framework for evaluating our models (Patil et al., 2024).

For SWE-Bench evaluation, we run our compressed models with the mini-SWE-agent scaffolding (Yang et al., 2024b) and report the score on the SWE-Bench Verified test set (Neil Chowdhury et al., 2024). We use 4,096 and 16,384 as the maximum number of output tokens for evaluating Qwen3-Coder-480B and Kimi-K2-Instruct-W4A16 on SWE-Bench, respectively. The input context length for both models is limited to 65,536. We do not limit the number of turns in mini-SWE-agent flow, but restart the rollout in cases where the model could not generate a valid patch (that is, in the case when the output of the final turn does not contain a `diff --git` substring). We set the maximum number of restarts to 20, which we found to be sufficient to generate patches for all samples with pruned models, unless the model produces degenerate responses like repeating strings. We use the cloud-based evaluation provided with the `sb-cli` tool to get the final scores for all evaluated models.

For $\tau^2$-bench Barres et al. (2025), we use greedy decoding and 4,096 as the maximum number of output tokens for each LLM call. For user simulation, we use the `gpt-4.1-2025-04-14` model;

---

[1]Reported as the `acc_norm` field in the EleutherAI evaluation harness outputs. See Gao (2021) for details.

maximum number of steps is 100 and number of trials is set to three for each domain. Following Artificial Analysis (2025), we additionally implement an LLM-based repetition checking step. Every 30 steps of the simulation, a model (in our case, `gpt-4.1-mini-2025-04-14`) is given the past 30 *episodes* of the conversation trajectory with a repetition checking prompt to determine whether the agent is stuck in the loop or making meaningful progress. This allows early task termination if the agent is stuck. We use the same decoding parameters for the repetition model as for the user and assistant models.

**Math and creative writing evaluation.** Mathematical reasoning is assessed on GSM8K (Cobbe et al., 2021) and MATH-500 (Hendrycks et al., 2021b; Lightman et al., 2023) benchmarks using the evalscope (ModelScope Team, 2024) framework. To assess creative writing, we use 146 creative writing prompts sampled from WildBench (Lin et al., 2024a) with `gpt-4o-2024-05-13` used as the judge to evaluate the model responses. We report normalized scores using the WildBench rubric.

**Generation configuration.** For models with $\leq$ 110B parameters, we use greedy sampling (i.e, temperature = 0.0) to evaluate code generation and math reasoning. For creative writing we use the default temperature, top-P, and top-K settings for each respective model. The maximum number of output tokens is extended to 16,384 for all generative tasks to account for the verbosity of some models. For hybrid reasoning models such as Qwen3-30B-A3B, we disable reasoning on all tasks by setting `enable_thinking=False` in the chat template.

For larger models with $\geq$ 110B parameters, we use greedy sampling for EvalPlus, SWE-Bench, and BFCLv3. On LiveCodeBench, Qwen3-Coder-480B and Kimi-K2 are evaluated with default sampling parameters and greedy sampling, respectively. We report the mean and standard deviation for Qwen3-Coder-480B on LiveCodeBench over five random seeds. We use a repetition penalty of 1.05 for all large model evaluations. For EvalPlus we use 768 as the maximum number of output tokens and 16,384 for LiveCodeBench. For BFCLv3 we set the maximum number of output tokens to 4,096.

**Model details.** The Kimi-K2-Instruct-W4A16 model used throughout this study is an INT4 weight-quantized version of Kimi-K2-Instruct released by RedHatAI (2025).

## D  BASELINE METHODS

The following formally describes the baselines compression methods we consider.

**Notation.** Let $\mathcal{X}_{cal}$ be a calibration dataset. Consider a SMoE model with $n$ layers, $L_n$, $K$ experts per layer $f_1, ... , f_K$, each a function $f_k : \mathbb{R}^d \to \mathbb{R}^d$, and a router producing non-negative gates $\mathbf{g}(x) = (g_1(x),...,g_K(x)) \in \Delta^{K-1}$. The output of layer $L_n$ is

$$h_n = \sum_i^K g_i(x) f_i(x).$$

The expert usage frequency, $\nu_i$, for expert $f_i$ is the number of tokens in $\mathcal{X}_{cal}$ for which $f_i$ is activated

$$\nu_i = |\mathcal{X}_i|,$$

where $\mathcal{X}_i = \{x \in \mathcal{X}_{cal} \,|\, i \in \text{TopK}(\mathbf{g}(x))\}$.

Given saliency scores, $\mathbf{S} \in \mathbb{R}^K$, pruning removes experts with the minimum saliency score. For merging, we first cluster experts based on their pairwise distances, $\mathbf{D} \in \mathbb{R}^{K \times K}$, and then merge the parameters of experts contained within each cluster.

**Frequency-based pruning.** The frequency-based pruning saliency criterion prunes experts with the lowest usage frequency across the calibration dataset. The saliency of $f_i$ is simply $S_i = \nu_i$.

**EAN pruning.** EAN pruning introduced by Jaiswal et al. (2025) accumulates the activation norm of each expert across tokens for which the expert is activated. The saliency of $f_i$ is

$$S_i = \sum_{x \in \mathcal{X}_i} \|f_i(x)\|_2. \tag{13}$$

**M-SMoE merging.** Proposed by Li et al. (2023), M-SMoE first uses weight-matching (Ainsworth et al., 2023) to find a permutation matrix $\mathbf{P_j}$ which aligns expert $f_j$ to expert $f_i$. In the models we study, each expert is a two-layer feed-forward SwiGLU block (Shazeer, 2020) with up, gate, and down projections: $f_j = \{W_{up}^{(j)}, W_{gate}^{(j)}, W_{down}^{(j)}\}$. The permutation matrix is applied to the intermediate dimension of the experts such that the expert outputs are invariant to the transformation

$$W_{up}'^{(j)} = W_{up}^{(j)}\mathbf{P}_j, \qquad W_{gate}'^{(j)} = W_{gate}^{(j)}\mathbf{P}_j, \qquad W_{down}'^{(j)} = \mathbf{P}_j^T W_{down}^{(j)}.$$

The permuted expert is defined as $\tilde{f}_j = \{W_{up}'^{(j)}, W_{gate}'^{(j)}, W_{down}'^{(j)}\}$.

To initialize the expert clusters, M-SMoE identifies the set of $m$ *dominant* experts $\mathbb{F}_{dom}$, as the experts across all layers with the highest usage frequency $\nu$. The pairwise expert distance is based on the cosine distance of the router gate-values measured on the calibration dataset

$$D_{i,j} = \frac{1}{|\mathcal{X}_{cal}|}\sum_{x\in\mathcal{X}_{cal}} 1 - \frac{g_i(x)\cdot g_j(x)}{\|g_i(x)\|\|g_j(x)\|}. \tag{14}$$

Non-dominant expert $j$ is clustered by selecting the dominant expert with the smallest pairwise distance

$$i^* = \underset{i\in\mathbb{F}_{dom}}{\arg\min} D_{i,j}.$$

The merged expert $f_\alpha$ is created by calculating the frequency-weighted average of the permuted parameters, $W'$, of all experts in the cluster $\mathbb{C}_\alpha$

$$\tilde{W}_a = \frac{\sum_{i\in\mathbb{C}_\alpha}\nu_i W_i'}{\sum_{i\in\mathbb{C}_\alpha}\nu_i}. \tag{15}$$

**HC-SMoE merging.** Chen et al. (2025) clusters experts based on their *representative vectors*, $A_i$, defined as the average activation across every token in the calibration dataset

$$A_i := \mathbb{E}_{x\sim\mathcal{X}_{cal}}[f_i(x)] = \frac{1}{|\mathcal{X}_{cal}|}\sum_{x\in\mathcal{X}_{cal}} f_i(x).$$

The expert pairwise distance is defined as the cosine distance between representative vectors

$$D_{i,j} = 1 - \frac{A_i\cdot A_j}{\|A_i\|\|A_j\|}. \tag{16}$$

Clusters are formed using hierarchical agglomerative clustering with average linkage criterion. We start by initializing each expert as a singleton cluster. At every iteration, the closest pair of clusters, $\mathbb{C}_i^*, \mathbb{C}_j^*$ are joined and the pairwise distances updated as the average of the constituents

$$i^*, j^* = \underset{i,j}{\arg\min} D_{i,j}, \qquad \mathbb{C}_\alpha = \mathbb{C}_{i^*}\cup\mathbb{C}_{j^*}, \qquad D_{a,k} = \frac{\sum_{i\in\mathbb{C}_\alpha}D_{i,k}}{|\mathbb{C}_\alpha|}.$$

The clusters are merged with Equation (15).

## E  QUANTIZATION AND EXPERT PRUNING

Pruning is easily combined with quantization, as demonstrated by our large-scale evaluations. We calibrate, prune, and evaluate Qwen3-Coder-480B and Kimi-K2 using FP8 and W4A16 quantization, respectively. In contrast, combining expert merging with quantization is more complex; the fine-grained shared quantization scales across parameter groups must be reconciled during merging, whereas pruning requires no such adjustment

Generally, weight quantization offers superior accuracy at a given compression ratio down to 4-bits. For instance, quantizing Qwen3-30B-A3B to 4-bits using AWQ (Lin et al., 2024b) outperforms a 50% expert-pruned REAP model at 16-bits, despite having half the checkpoint size. However, combining these methods enables compression rates that neither can achieve in isolation. For example, our Kimi-K2 results combine 4-bit weights with 50% expert pruning, representing a total size reduction of 87.5%. Achieving a

similar footprint through quantization alone would require 2-bit weights, which are not natively supported on most current accelerators and typically suffer from significant accuracy degradation (Liu et al., 2025b).

To further illustrate this, we calibrate and quantize Qwen3-30B-A3B using the evol-codealpaca-v1 dataset to 4- and 2-bit weights with the LLM Compressor library (Red Hat AI and vLLM Project, 2024). We additionally compress the 4-bit checkpoint by using REAP calibrated on the same dataset to prune half the experts. In Table A5, we report EvalPlus accuracy versus checkpoint size relative to the uncompressed 16-bit model. This preliminary analysis suggests that REAP combined with 4-bit weights outperforms more aggressive quantization. A more thorough exploration of various low-bit schemes and pruning combinations remains an important direction for future work.

| Quantization | Num. Experts | Approx. Relative Size | Eval+ |
|---|---|---|---|
| W16A16 | 128 | 100.0% | 81.4 |
| W16A16 | 64 | 50.0% | 78.0 |
| W4A16-G128 | 128 | 25.0% | 80.5 |
| W4A16-G128 | 64 | 12.5% | 77.6 |
| W2A16-G128 | 64 | 12.5% | 28.6 |

Table A5: **Qwen3-30B-A3B EvalPlus accuracy vs. relative checkpoint size** with AWQ quantization, REAP expert pruning, and quantization combined with expert pruning. Overall, quantization outperforms expert pruning up to 4-bit weights. Beyond 4-bit weights, combining quantization with expert pruning yields significantly better accuracy than more aggressive quantization.

## F  ADDITIONAL RESULTS

Table A6 shows the full suite of MC question answering benchmarks and the average result across all models and methods. Table A7 tabulates code generation accuracy of compressed SMoE models calibrated on evol-codealpaca. Eval+ is the average of MBPP+ and HE+. The *Code Avg* column is the average of Eval+ and LiveCodeBench (LiveCode). Table A8 summarizes the accuracy of the various compression methods studied when calibrated with the C4 dataset on coding and MC benchmarks. Notably, while the MC performance is generally slightly higher than models calibrated on evol-codealpaca, the resulting code generation quality is abysmal, with most models failing to generate coherent output.

Figure A6 plots non-agentic coding and MC accuracy versus compressed model size. Figure A7a depicts the proportion of singleton clusters for HC-SMoE and M-SMoE. Figure A7b plots accuracy vs. maximum cluster sizes when the maximum cardinality of clusters is restricted. Figures A8 and A9 show the importance of using domain-specific calibration data, particularly at high compression ratios.

Table A9 presents the complete $\tau^2$-bench results across three domains (Retail, Airline, and Telecom) for the baseline model and REAP compression at 25% and 50% levels. The results show pass^k metrics for k=1, 2, and 3, demonstrating the impact of pruning on evaluating conversational agents, specifically designed to test their ability to collaborate with a user in real-world scenarios.

Table A10 presents additional $\tau^2$-bench results with REAP compression for a set of non-reasoning and reasoning models (Qwen3-Coder-30B, GLM4.5-Air, MiniMax-M2). Furthermore, Table A11 presents BFCLv3 scores for the same set of REAP-compressed models (with an inclusion of REAP-compressed GLM-4.6-FP8 variants in thinking mode). The calibration data mix used for non-reasoning models in these experiments is the same as for the other large-scale models in this study (12,228 samples including tool calling and agentic trajectory data). For the models that support reasoning, we include an extra 12,228 samples from the Open-R1 Mixture-of-Thoughts dataset (HuggingFace, 2025), drawing 4,096 samples from the Code, Math and Science subsets with a maximum sequence length of 16,384 tokens.

Table A12 shows results for the Kimi-Linear-48B-A3B-Instruct model as an example of REAP compression applied to a hybrid linear attention architecture. The calibration data mix is made up of 24,414 samples with a maximum sequence length of 16,384 tokens. The samples are drawn from evol-codealpaca (Chaudhary, 2023; Luo et al., 2024; Tam, 2023), xlam-function-calling (Liu et al., 2024c; Salesforce, 2025), SWE-smith-trajectories (Yang et al., 2025c;b), WildChat (Zhao et al., 2024), NuminaMath (LI et al., 2024) and Aya (Singh et al., 2024). The calibration data mix composition reflects coding, tool calling, multilingual and general conversational use cases. We evaluate Kimi-Linear-48B-A3B-Instruct on coding and math benchmarks at 30% REAP compression, as well as on long-context question-answering

Table A6: Detailed benchmark results for multiple-choice QA tasks.

| Model | Compression | Technique | Method | ARC-c | ARC-e | BoolQ | Hellaswag | MMLU | OBQA | RTE | WinoG. | MC Avg |
|---|---|---|---|---|---|---|---|---|---|---|---|---|
| ERNIE-4.5-21B-A3B-PT | Baseline | | | 0.564 | 0.782 | 0.873 | 0.813 | 0.737 | 0.462 | 0.812 | 0.724 | 0.721 |
| | 25% | Merging | M-SMoE | 0.434 ± 0.006 | 0.652 ± 0.008 | 0.846 ± 0.001 | 0.597 ± 0.002 | 0.591 ± 0.001 | 0.350 ± 0.006 | 0.819 ± 0.010 | 0.655 ± 0.003 | 0.618 ± 0.002 |
| | | | HC-SMoE | 0.506 ± 0.000 | 0.717 ± 0.001 | 0.849 ± 0.001 | 0.714 ± 0.001 | 0.652 ± 0.002 | 0.371 ± 0.002 | 0.799 ± 0.002 | 0.674 ± 0.004 | 0.660 ± 0.001 |
| | | Pruning | Frequency | 0.486 ± 0.004 | 0.711 ± 0.000 | 0.852 ± 0.004 | 0.675 ± 0.003 | 0.628 ± 0.003 | 0.373 ± 0.003 | 0.780 ± 0.006 | 0.676 ± 0.005 | 0.648 ± 0.001 |
| | | | EAN | 0.498 ± 0.005 | 0.713 ± 0.002 | 0.863 ± 0.002 | 0.717 ± 0.004 | 0.625 ± 0.001 | 0.405 ± 0.011 | 0.811 ± 0.009 | 0.702 ± 0.005 | **0.667 ± 0.000** |
| | | | REAP | 0.526 ± 0.004 | 0.756 ± 0.006 | 0.858 ± 0.003 | 0.704 ± 0.001 | 0.636 ± 0.002 | 0.401 ± 0.001 | 0.765 ± 0.010 | 0.690 ± 0.002 | **0.667 ± 0.000** |
| | 50% | Merging | M-SMoE | 0.294 ± 0.033 | 0.452 ± 0.040 | 0.764 ± 0.010 | 0.341 ± 0.017 | 0.385 ± 0.001 | 0.270 ± 0.004 | 0.687 ± 0.017 | 0.529 ± 0.010 | 0.465 ± 0.012 |
| | | | HC-SMoE | 0.411 ± 0.003 | 0.641 ± 0.002 | 0.822 ± 0.001 | 0.523 ± 0.001 | 0.495 ± 0.002 | 0.330 ± 0.005 | 0.742 ± 0.011 | 0.587 ± 0.009 | 0.569 ± 0.001 |
| | | Pruning | Frequency | 0.400 ± 0.002 | 0.584 ± 0.006 | 0.830 ± 0.001 | 0.522 ± 0.003 | 0.506 ± 0.006 | 0.303 ± 0.004 | 0.758 ± 0.004 | 0.625 ± 0.004 | 0.566 ± 0.002 |
| | | | EAN | 0.417 ± 0.005 | 0.633 ± 0.005 | 0.830 ± 0.003 | 0.572 ± 0.001 | 0.509 ± 0.002 | 0.336 ± 0.003 | 0.785 ± 0.014 | 0.626 ± 0.003 | **0.589 ± 0.003** |
| | | | REAP | 0.407 ± 0.003 | 0.628 ± 0.006 | 0.820 ± 0.002 | 0.551 ± 0.004 | 0.491 ± 0.001 | 0.331 ± 0.007 | 0.767 ± 0.004 | 0.614 ± 0.002 | 0.576 ± 0.001 |
| Qwen3-30B-A3B | Baseline | | | 0.563 | 0.790 | 0.887 | 0.778 | 0.779 | 0.454 | 0.816 | 0.702 | 0.721 |
| | 25% | Merging | M-SMoE | 0.357 ± 0.006 | 0.519 ± 0.003 | 0.843 ± 0.006 | 0.529 ± 0.002 | 0.536 ± 0.004 | 0.310 ± 0.005 | 0.735 ± 0.027 | 0.635 ± 0.005 | 0.558 ± 0.003 |
| | | | HC-SMoE | 0.478 ± 0.006 | 0.722 ± 0.006 | 0.863 ± 0.003 | 0.714 ± 0.000 | 0.684 ± 0.002 | 0.417 ± 0.001 | 0.805 ± 0.004 | 0.710 ± 0.004 | **0.674 ± 0.001** |
| | | Pruning | Frequency | 0.401 ± 0.011 | 0.600 ± 0.016 | 0.847 ± 0.003 | 0.593 ± 0.003 | 0.600 ± 0.004 | 0.342 ± 0.012 | 0.781 ± 0.002 | 0.637 ± 0.005 | 0.600 ± 0.005 |
| | | | EAN | 0.406 ± 0.007 | 0.603 ± 0.014 | 0.847 ± 0.005 | 0.607 ± 0.006 | 0.600 ± 0.002 | 0.337 ± 0.003 | 0.764 ± 0.002 | 0.660 ± 0.009 | 0.603 ± 0.003 |
| | | | REAP | 0.481 ± 0.004 | 0.727 ± 0.002 | 0.855 ± 0.004 | 0.700 ± 0.006 | 0.673 ± 0.001 | 0.399 ± 0.008 | 0.789 ± 0.014 | 0.696 ± 0.003 | 0.665 ± 0.002 |
| | 50% | Merging | M-SMoE | 0.278 ± 0.003 | 0.402 ± 0.003 | 0.753 ± 0.004 | 0.399 ± 0.002 | 0.366 ± 0.004 | 0.278 ± 0.002 | 0.586 ± 0.014 | 0.546 ± 0.004 | 0.451 ± 0.002 |
| | | | HC-SMoE | 0.368 ± 0.002 | 0.593 ± 0.003 | 0.740 ± 0.003 | 0.473 ± 0.002 | 0.516 ± 0.003 | 0.301 ± 0.007 | 0.724 ± 0.004 | 0.620 ± 0.005 | **0.542 ± 0.001** |
| | | Pruning | Frequency | 0.285 ± 0.001 | 0.424 ± 0.002 | 0.779 ± 0.003 | 0.458 ± 0.003 | 0.397 ± 0.002 | 0.286 ± 0.004 | 0.659 ± 0.012 | 0.570 ± 0.009 | 0.483 ± 0.001 |
| | | | EAN | 0.296 ± 0.006 | 0.426 ± 0.009 | 0.759 ± 0.007 | 0.471 ± 0.002 | 0.443 ± 0.001 | 0.291 ± 0.009 | 0.668 ± 0.020 | 0.589 ± 0.009 | 0.493 ± 0.003 |
| | | | REAP | 0.354 ± 0.006 | 0.503 ± 0.008 | 0.737 ± 0.009 | 0.481 ± 0.004 | 0.496 ± 0.003 | 0.309 ± 0.001 | 0.561 ± 0.020 | 0.584 ± 0.004 | 0.503 ± 0.002 |
| Mixtral-8x7B-Instruct-v0.1 | Baseline | | | 0.650 | 0.842 | 0.887 | 0.861 | 0.691 | 0.496 | 0.722 | 0.740 | 0.736 |
| | 25% | Merging | M-SMoE | 0.532 ± 0.004 | 0.769 ± 0.007 | 0.847 ± 0.001 | 0.747 ± 0.002 | 0.553 ± 0.001 | 0.429 ± 0.008 | 0.632 ± 0.010 | 0.656 ± 0.004 | 0.646 ± 0.001 |
| | | | HC-SMoE | 0.590 ± 0.004 | 0.797 ± 0.004 | 0.869 ± 0.003 | 0.835 ± 0.002 | 0.626 ± 0.000 | 0.482 ± 0.004 | 0.703 ± 0.012 | 0.731 ± 0.007 | 0.704 ± 0.001 |
| | | Pruning | Frequency | 0.616 ± 0.014 | 0.826 ± 0.007 | 0.875 ± 0.001 | 0.825 ± 0.002 | 0.637 ± 0.003 | 0.451 ± 0.003 | 0.706 ± 0.017 | 0.692 ± 0.005 | 0.704 ± 0.002 |
| | | | EAN | 0.607 ± 0.003 | 0.831 ± 0.001 | 0.884 ± 0.001 | 0.836 ± 0.001 | 0.646 ± 0.002 | 0.484 ± 0.005 | 0.700 ± 0.004 | 0.732 ± 0.004 | **0.715 ± 0.000** |
| | | | REAP | 0.600 ± 0.004 | 0.822 ± 0.001 | 0.872 ± 0.000 | 0.830 ± 0.000 | 0.642 ± 0.000 | 0.469 ± 0.002 | 0.771 ± 0.002 | 0.716 ± 0.002 | **0.715 ± 0.000** |
| | 50% | Merging | M-SMoE | 0.446 ± 0.005 | 0.700 ± 0.001 | 0.788 ± 0.003 | 0.630 ± 0.002 | 0.430 ± 0.001 | 0.386 ± 0.003 | 0.570 ± 0.000 | 0.596 ± 0.003 | 0.568 ± 0.001 |
| | | | HC-SMoE | 0.539 ± 0.003 | 0.759 ± 0.000 | 0.851 ± 0.001 | 0.791 ± 0.001 | 0.543 ± 0.000 | 0.442 ± 0.000 | 0.700 ± 0.004 | 0.712 ± 0.002 | 0.667 ± 0.001 |
| | | Pruning | Frequency | 0.541 ± 0.004 | 0.781 ± 0.003 | 0.824 ± 0.013 | 0.759 ± 0.002 | 0.516 ± 0.002 | 0.411 ± 0.006 | 0.708 ± 0.023 | 0.650 ± 0.005 | 0.649 ± 0.004 |
| | | | EAN | 0.551 ± 0.014 | 0.774 ± 0.008 | 0.859 ± 0.004 | 0.794 ± 0.002 | 0.550 ± 0.006 | 0.452 ± 0.014 | 0.717 ± 0.023 | 0.693 ± 0.008 | **0.674 ± 0.005** |
| | | | REAP | 0.546 ± 0.006 | 0.782 ± 0.003 | 0.841 ± 0.002 | 0.777 ± 0.001 | 0.565 ± 0.000 | 0.458 ± 0.003 | 0.739 ± 0.008 | 0.677 ± 0.005 | 0.673 ± 0.001 |
| Llama-4-Scout-17B-16E-Instruct | Baseline | | | 0.627 | 0.848 | 0.879 | 0.823 | 0.803 | 0.462 | 0.765 | 0.692 | 0.738 |
| | 25% | Merging | M-SMoE | 0.573 | 0.802 | 0.872 | 0.752 | 0.719 | 0.434 | 0.769 | 0.671 | 0.699 |
| | | | HC-SMoE | 0.588 | 0.814 | 0.876 | 0.779 | 0.720 | 0.424 | 0.729 | 0.695 | 0.703 |
| | | Pruning | Frequency | 0.584 | 0.817 | 0.876 | 0.779 | 0.733 | 0.438 | 0.773 | 0.691 | 0.711 |
| | | | EAN | 0.582 | 0.816 | 0.872 | 0.777 | 0.735 | 0.446 | 0.791 | 0.679 | 0.712 |
| | | | REAP | 0.594 | 0.830 | 0.872 | 0.788 | 0.756 | 0.452 | 0.769 | 0.683 | **0.718** |
| | 50% | Merging | M-SMoE | 0.498 | 0.717 | 0.856 | 0.676 | 0.609 | 0.388 | 0.787 | 0.665 | 0.649 |
| | | | HC-SMoE | 0.526 | 0.781 | 0.862 | 0.718 | 0.628 | 0.386 | 0.726 | 0.660 | 0.661 |
| | | Pruning | Frequency | 0.518 | 0.734 | 0.860 | 0.704 | 0.652 | 0.398 | 0.765 | 0.657 | 0.661 |
| | | | EAN | 0.510 | 0.750 | 0.857 | 0.712 | 0.650 | 0.398 | 0.762 | 0.662 | 0.663 |
| | | | REAP | 0.561 | 0.802 | 0.869 | 0.745 | 0.682 | 0.432 | 0.762 | 0.664 | **0.689** |
| GLM-4.5-Air | Baseline | | | 0.619 | 0.825 | 0.882 | 0.858 | 0.789 | 0.478 | 0.747 | 0.776 | 0.747 |
| | 25% | Merging | M-SMoE | 0.429 | 0.651 | 0.808 | 0.671 | 0.578 | 0.362 | 0.578 | 0.695 | 0.596 |
| | | | HC-SMoE | 0.577 | 0.782 | 0.860 | 0.815 | 0.722 | 0.458 | 0.668 | 0.755 | **0.704** |
| | | Pruning | Frequency | 0.493 | 0.715 | 0.827 | 0.732 | 0.653 | 0.422 | 0.614 | 0.725 | 0.648 |
| | | | EAN | 0.492 | 0.705 | 0.805 | 0.736 | 0.656 | 0.368 | 0.603 | 0.730 | 0.637 |
| | | | REAP | 0.565 | 0.758 | 0.809 | 0.793 | 0.701 | 0.430 | 0.610 | 0.725 | 0.674 |
| | 50% | Merging | M-SMoE | 0.291 | 0.452 | 0.693 | 0.433 | 0.382 | 0.266 | 0.484 | 0.551 | 0.444 |
| | | | HC-SMoE | 0.428 | 0.671 | 0.761 | 0.590 | 0.524 | 0.318 | 0.603 | 0.613 | **0.564** |
| | | Pruning | Frequency | 0.334 | 0.535 | 0.767 | 0.566 | 0.478 | 0.288 | 0.567 | 0.635 | 0.521 |
| | | | EAN | 0.358 | 0.530 | 0.682 | 0.573 | 0.489 | 0.300 | 0.516 | 0.635 | 0.511 |
| | | | REAP | 0.418 | 0.586 | 0.684 | 0.628 | 0.550 | 0.312 | 0.581 | 0.673 | 0.554 |
| Qwen3-Coder-480B-A35B-Instruct-FP8 | Baseline | | | 0.644 | 0.822 | 0.906 | 0.841 | 0.850 | 0.468 | 0.751 | 0.717 | 0.750 |
| | 25% | Pruning | Frequency | 0.443 | 0.673 | 0.845 | 0.651 | 0.621 | 0.280 | 0.704 | 0.632 | 0.606 |
| | | | EAN | 0.555 | 0.766 | 0.891 | 0.769 | 0.795 | 0.404 | 0.747 | 0.691 | 0.702 |
| | | | REAP | 0.635 | 0.824 | 0.900 | 0.841 | 0.836 | 0.466 | 0.754 | 0.725 | **0.748** |
| | 50% | Pruning | Frequency | 0.314 | 0.470 | 0.791 | 0.502 | 0.451 | 0.262 | 0.679 | 0.580 | 0.506 |
| | | | EAN | 0.402 | 0.596 | 0.858 | 0.629 | 0.615 | 0.216 | 0.744 | 0.666 | 0.591 |
| | | | REAP | 0.546 | 0.772 | 0.872 | 0.756 | 0.696 | 0.430 | 0.762 | 0.701 | **0.692** |
| Kimi-K2-Instruct-W4A16 | Baseline | | | 0.712 | 0.879 | 0.913 | 0.765 | 0.872 | 0.504 | 0.783 | 0.811 | 0.780 |
| | 25% | Pruning | Frequency | 0.518 | 0.771 | 0.825 | 0.787 | 0.242 | 0.420 | 0.653 | 0.613 | 0.604 |
| | | | EAN | 0.615 | 0.819 | 0.893 | 0.843 | 0.500 | 0.446 | 0.762 | 0.743 | 0.703 |
| | | | REAP | 0.671 | 0.854 | 0.907 | 0.860 | 0.809 | 0.470 | 0.805 | 0.809 | **0.773** |
| | 50% | Pruning | Frequency | 0.285 | 0.498 | 0.620 | 0.436 | 0.241 | 0.314 | 0.617 | 0.500 | 0.439 |
| | | | EAN | 0.426 | 0.682 | 0.863 | 0.663 | 0.324 | 0.356 | 0.726 | 0.659 | 0.587 |
| | | | REAP | 0.476 | 0.661 | 0.883 | 0.643 | 0.636 | 0.350 | 0.816 | 0.681 | **0.643** |

benchmarks LongBench v2 (Bai et al., 2025) and FRAMES (Krishna et al., 2024). For long-context benchmarks, a truncation in the middle is done for input prompts exceeding 131,072 tokens in length. The accuracy scores for FRAMES are estimated with LLM-as-a-judge (`gpt-4.1-2025-04-14` model as judge). Notably, even though the maximum sequence length during calibration is 16,384 tokens, the REAP-compressed model retains performance at context lengths far exceeding that value.

Table A7: Detailed benchmark results for non-agentic code generation tasks. Eval+ is the average of MBPP+ and HE+. The Code Avg column is the average of Eval+ and LiveCodeBench (LiveCode).

| Model | Compression | Technique | Method | HE | HE+ | MBPP | MBPP+ | Eval+ | LiveCode | Code Avg |
|---|---|---|---|---|---|---|---|---|---|---|
| ERNIE-4.5-21B-A3B-PT | Baseline | | | 0.902 | 0.866 | 0.910 | 0.765 | 0.815 | 0.231 | 0.523 |
| | 25% | Merging | M-SMoE | 0.774 ± 0.011 | 0.730 ± 0.009 | 0.768 ± 0.015 | 0.647 ± 0.017 | 0.688 ± 0.007 | 0.194 ± 0.022 | 0.441 ± 0.011 |
| | | | HC-SMoE | 0.837 ± 0.007 | 0.805 ± 0.000 | 0.827 ± 0.003 | 0.696 ± 0.008 | 0.750 ± 0.004 | 0.207 ± 0.008 | 0.479 ± 0.003 |
| | | Pruning | Frequency | 0.890 ± 0.006 | 0.846 ± 0.009 | 0.837 ± 0.010 | 0.709 ± 0.010 | 0.777 ± 0.009 | 0.151 ± 0.096 | 0.464 ± 0.044 |
| | | | EAN | 0.890 ± 0.006 | 0.848 ± 0.011 | 0.840 ± 0.006 | 0.727 ± 0.004 | 0.787 ± 0.007 | 0.161 ± 0.111 | 0.474 ± 0.053 |
| | | | REAP | 0.909 ± 0.012 | 0.868 ± 0.004 | 0.866 ± 0.006 | 0.735 ± 0.002 | 0.801 ± 0.002 | 0.223 ± 0.008 | **0.512 ± 0.005** |
| | 50% | Merging | M-SMoE | 0.104 ± 0.022 | 0.100 ± 0.029 | 0.239 ± 0.036 | 0.207 ± 0.040 | 0.153 ± 0.015 | 0.024 ± 0.008 | 0.089 ± 0.009 |
| | | | HC-SMoE | 0.425 ± 0.004 | 0.404 ± 0.007 | 0.608 ± 0.018 | 0.511 ± 0.011 | 0.458 ± 0.008 | 0.082 ± 0.015 | 0.270 ± 0.008 |
| | | Pruning | Frequency | 0.699 ± 0.031 | 0.640 ± 0.022 | 0.696 ± 0.014 | 0.584 ± 0.006 | 0.612 ± 0.014 | 0.083 ± 0.066 | 0.348 ± 0.026 |
| | | | EAN | 0.675 ± 0.019 | 0.642 ± 0.009 | 0.713 ± 0.015 | 0.591 ± 0.016 | 0.617 ± 0.012 | 0.112 ± 0.064 | 0.364 ± 0.034 |
| | | | REAP | 0.787 ± 0.038 | 0.752 ± 0.035 | 0.749 ± 0.005 | 0.638 ± 0.013 | 0.695 ± 0.024 | 0.187 ± 0.005 | **0.441 ± 0.014** |
| Qwen3-30B-A3B | Baseline | | | 0.927 | 0.884 | 0.881 | 0.743 | 0.814 | 0.302 | 0.558 |
| | 25% | Merging | M-SMoE | 0.878 ± 0.012 | 0.833 ± 0.007 | 0.849 ± 0.007 | 0.728 ± 0.007 | 0.781 ± 0.007 | 0.293 ± 0.017 | 0.537 ± 0.006 |
| | | | HC-SMoE | 0.866 ± 0.011 | 0.805 ± 0.016 | 0.832 ± 0.006 | 0.698 ± 0.005 | 0.752 ± 0.006 | 0.258 ± 0.006 | 0.505 ± 0.003 |
| | | Pruning | Frequency | 0.921 ± 0.006 | 0.874 ± 0.007 | 0.868 ± 0.000 | 0.735 ± 0.003 | 0.805 ± 0.005 | 0.302 ± 0.011 | 0.553 ± 0.003 |
| | | | EAN | 0.909 ± 0.006 | 0.864 ± 0.004 | 0.859 ± 0.009 | 0.729 ± 0.008 | 0.797 ± 0.005 | 0.311 ± 0.018 | **0.554 ± 0.010** |
| | | | REAP | 0.911 ± 0.004 | 0.870 ± 0.004 | 0.847 ± 0.004 | 0.725 ± 0.008 | 0.797 ± 0.004 | 0.304 ± 0.003 | 0.551 ± 0.004 |
| | 50% | Merging | M-SMoE | 0.687 ± 0.013 | 0.638 ± 0.004 | 0.618 ± 0.004 | 0.541 ± 0.007 | 0.590 ± 0.005 | 0.205 ± 0.019 | 0.397 ± 0.007 |
| | | | HC-SMoE | 0.577 ± 0.023 | 0.541 ± 0.013 | 0.631 ± 0.010 | 0.546 ± 0.004 | 0.543 ± 0.005 | 0.185 ± 0.018 | 0.364 ± 0.007 |
| | | Pruning | Frequency | 0.787 ± 0.016 | 0.756 ± 0.022 | 0.692 ± 0.016 | 0.579 ± 0.016 | 0.668 ± 0.019 | 0.236 ± 0.025 | 0.452 ± 0.022 |
| | | | EAN | 0.886 ± 0.025 | 0.837 ± 0.020 | 0.798 ± 0.006 | 0.669 ± 0.008 | 0.753 ± 0.011 | 0.306 ± 0.032 | 0.530 ± 0.004 |
| | | | REAP | 0.917 ± 0.013 | 0.858 ± 0.015 | 0.818 ± 0.008 | 0.703 ± 0.004 | 0.780 ± 0.006 | 0.302 ± 0.000 | **0.541 ± 0.003** |
| Mixtral-8x7B-Instruct-v0.1 | Baseline | | | 0.524 | 0.476 | 0.556 | 0.463 | 0.469 | 0.123 | 0.296 |
| | 25% | Merging | M-SMoE | 0.315 ± 0.007 | 0.270 ± 0.015 | 0.446 ± 0.007 | 0.380 ± 0.015 | 0.325 ± 0.015 | 0.033 ± 0.010 | 0.179 ± 0.011 |
| | | | HC-SMoE | 0.439 ± 0.028 | 0.386 ± 0.020 | 0.530 ± 0.022 | 0.441 ± 0.007 | 0.414 ± 0.007 | 0.110 ± 0.010 | 0.262 ± 0.001 |
| | | Pruning | Frequency | 0.400 ± 0.034 | 0.358 ± 0.035 | 0.541 ± 0.006 | 0.453 ± 0.012 | 0.405 ± 0.019 | 0.099 ± 0.014 | 0.252 ± 0.004 |
| | | | EAN | 0.413 ± 0.027 | 0.366 ± 0.024 | 0.477 ± 0.009 | 0.409 ± 0.013 | 0.388 ± 0.015 | 0.111 ± 0.006 | 0.249 ± 0.006 |
| | | | REAP | 0.445 ± 0.016 | 0.388 ± 0.025 | 0.548 ± 0.010 | 0.470 ± 0.011 | 0.429 ± 0.008 | 0.097 ± 0.006 | **0.263 ± 0.007** |
| | 50% | Merging | M-SMoE | 0.085 ± 0.026 | 0.076 ± 0.022 | 0.139 ± 0.121 | 0.118 ± 0.102 | 0.127 ± 0.011 | 0.004 ± 0.006 | 0.063 ± 0.005 |
| | | | HC-SMoE | 0.175 ± 0.015 | 0.146 ± 0.000 | 0.335 ± 0.026 | 0.282 ± 0.031 | 0.214 ± 0.015 | 0.013 ± 0.008 | 0.114 ± 0.007 |
| | | Pruning | Frequency | 0.187 ± 0.015 | 0.148 ± 0.007 | 0.342 ± 0.016 | 0.287 ± 0.012 | 0.217 ± 0.007 | 0.023 ± 0.004 | 0.120 ± 0.004 |
| | | | EAN | 0.220 ± 0.006 | 0.189 ± 0.006 | 0.375 ± 0.020 | 0.325 ± 0.015 | 0.257 ± 0.005 | 0.031 ± 0.011 | 0.144 ± 0.006 |
| | | | REAP | 0.258 ± 0.019 | 0.220 ± 0.016 | 0.381 ± 0.003 | 0.331 ± 0.008 | 0.275 ± 0.011 | 0.055 ± 0.010 | **0.165 ± 0.001** |
| Llama-4-Scout-17B-16E-Instruct | Baseline | | | 0.829 | 0.768 | 0.788 | 0.640 | 0.704 | 0.341 | 0.522 |
| | 25% | Merging | M-SMoE | 0.823 | 0.762 | 0.786 | 0.635 | 0.699 | 0.324 | 0.511 |
| | | | HC-SMoE | 0.787 | 0.738 | 0.735 | 0.587 | 0.663 | 0.148 | 0.405 |
| | | Pruning | Frequency | 0.835 | 0.768 | 0.788 | 0.630 | 0.699 | 0.317 | 0.508 |
| | | | EAN | 0.823 | 0.762 | 0.804 | 0.648 | 0.705 | 0.328 | **0.517** |
| | | | REAP | 0.829 | 0.787 | 0.788 | 0.622 | 0.704 | 0.242 | 0.473 |
| | 50% | Merging | M-SMoE | 0.787 | 0.732 | 0.762 | 0.614 | 0.673 | 0.187 | 0.430 |
| | | | HC-SMoE | 0.604 | 0.530 | 0.500 | 0.399 | 0.465 | 0.077 | 0.271 |
| | | Pruning | Frequency | 0.823 | 0.756 | 0.751 | 0.595 | 0.676 | 0.223 | 0.449 |
| | | | EAN | 0.805 | 0.744 | 0.754 | 0.601 | 0.672 | 0.209 | 0.441 |
| | | | REAP | 0.841 | 0.768 | 0.762 | 0.624 | 0.696 | 0.248 | **0.472** |
| GLM-4.5-Air | Baseline | | | 0.848 | 0.829 | 0.860 | 0.743 | 0.786 | 0.374 | 0.580 |
| | 25% | Merging | M-SMoE | 0.866 | 0.793 | 0.807 | 0.659 | 0.726 | 0.330 | 0.528 |
| | | | HC-SMoE | 0.872 | 0.805 | 0.825 | 0.669 | 0.737 | 0.363 | 0.550 |
| | | Pruning | Frequency | 0.848 | 0.811 | 0.854 | 0.706 | 0.759 | 0.341 | 0.550 |
| | | | EAN | 0.872 | 0.817 | 0.876 | 0.720 | 0.768 | 0.374 | 0.571 |
| | | | REAP | 0.884 | 0.829 | 0.839 | 0.688 | 0.759 | 0.412 | **0.585** |
| | 50% | Merging | M-SMoE | 0.518 | 0.500 | 0.519 | 0.437 | 0.468 | 0.099 | 0.284 |
| | | | HC-SMoE | 0.707 | 0.659 | 0.706 | 0.577 | 0.618 | 0.220 | 0.419 |
| | | Pruning | Frequency | 0.628 | 0.573 | 0.534 | 0.450 | 0.511 | 0.104 | 0.308 |
| | | | EAN | 0.841 | 0.780 | 0.807 | 0.661 | 0.721 | 0.253 | 0.487 |
| | | | REAP | 0.823 | 0.780 | 0.712 | 0.577 | 0.679 | 0.352 | **0.515** |
| Qwen3-Coder-480B-A35B-Instruct-FP8 | Baseline | | | 0.951 | 0.890 | 0.923 | 0.791 | 0.841 | 0.431 ± 0.011 | 0.636 |
| | 25% | Pruning | Frequency | 0.884 | 0.805 | 0.810 | 0.669 | 0.737 | 0.296 ± 0.017 | 0.516 |
| | | | EAN | 0.939 | 0.878 | 0.911 | 0.775 | 0.827 | 0.419 ± 0.015 | 0.623 |
| | | | REAP | 0.957 | 0.890 | 0.917 | 0.772 | 0.831 | 0.416 ± 0.013 | **0.624** |
| | 50% | Pruning | Frequency | 0.020 | 0.012 | 0.007 | 0.003 | 0.007 | 0.012 ± 0.001 | 0.010 |
| | | | EAN | 0.915 | 0.841 | 0.854 | 0.714 | 0.777 | 0.382 ± 0.012 | 0.580 |
| | | | REAP | 0.939 | 0.872 | 0.910 | 0.772 | 0.822 | 0.415 ± 0.015 | **0.619** |
| Kimi-K2-Instruct-W4A16 | Baseline | | | 0.963 | 0.921 | 0.913 | 0.735 | 0.828 | 0.434 | 0.631 |
| | 25% | Pruning | Frequency | 0.530 | 0.463 | 0.595 | 0.508 | 0.486 | 0.082 | 0.284 |
| | | | EAN | 0.909 | 0.860 | 0.857 | 0.698 | 0.779 | 0.379 | 0.579 |
| | | | REAP | 0.957 | 0.921 | 0.918 | 0.759 | 0.840 | 0.440 | **0.640** |
| | 50% | Pruning | Frequency | 0.098 | 0.079 | 0.175 | 0.146 | 0.112 | 0.000 | 0.056 |
| | | | EAN | 0.866 | 0.811 | 0.780 | 0.632 | 0.722 | 0.253 | 0.487 |
| | | | REAP | 0.915 | 0.884 | 0.899 | 0.754 | 0.819 | 0.429 | **0.624** |

Table A8: C4 calibrated results for coding and MC tasks.

| Model | Compression | Technique | Method | Eval+ | Coding LiveCode | Code Avg | ARC-c | ARC-e | BoolQ | Hellaswag | MC MMLU | OBQA | RTE | WinoG. | MC Avg |
|---|---|---|---|---|---|---|---|---|---|---|---|---|---|---|---|
| ERNIE-4.5-21B-A3B-PT | **Baseline** | | | 0.815 | 0.231 | 0.523 | 0.564 | 0.782 | 0.873 | 0.813 | 0.737 | 0.462 | 0.812 | 0.724 | 0.721 |
| | 25% | Merging | M-SMoE | 0.061 | 0.016 | 0.039 | 0.497 | 0.729 | 0.860 | 0.723 | 0.602 | 0.424 | 0.801 | 0.699 | 0.667 |
| | | | HC-SMoE | 0.369 | 0.099 | **0.234** | 0.515 | 0.728 | 0.860 | 0.745 | 0.649 | 0.428 | 0.794 | 0.694 | 0.677 |
| | | Pruning | Frequency | 0.254 | 0.000 | 0.127 | 0.515 | 0.735 | 0.841 | 0.719 | 0.588 | 0.382 | 0.791 | 0.683 | 0.657 |
| | | | EAN | 0.262 | 0.000 | 0.131 | 0.528 | 0.750 | 0.853 | 0.790 | 0.558 | 0.442 | 0.783 | 0.706 | 0.676 |
| | | | REAP | 0.212 | 0.055 | 0.133 | 0.553 | 0.784 | 0.843 | 0.775 | 0.635 | 0.454 | 0.798 | 0.708 | **0.694** |
| | 50% | Merging | M-SMoE | 0.000 | 0.000 | 0.000 | 0.297 | 0.460 | 0.674 | 0.449 | 0.312 | 0.280 | 0.671 | 0.575 | 0.465 |
| | | | HC-SMoE | 0.000 | 0.000 | 0.000 | 0.409 | 0.615 | 0.666 | 0.515 | 0.489 | 0.290 | 0.632 | 0.580 | 0.524 |
| | | Pruning | Frequency | 0.000 | 0.000 | 0.000 | 0.393 | 0.625 | 0.717 | 0.569 | 0.496 | 0.324 | 0.758 | 0.619 | 0.563 |
| | | | EAN | 0.007 | 0.003 | 0.005 | 0.451 | 0.676 | 0.742 | 0.687 | 0.474 | 0.398 | 0.736 | 0.691 | **0.607** |
| | | | REAP | 0.015 | 0.000 | **0.008** | 0.403 | 0.562 | 0.713 | 0.668 | 0.391 | 0.388 | 0.708 | 0.669 | 0.563 |
| Qwen3-30B-A3B | **Baseline** | | | 0.814 | 0.302 | 0.558 | 0.563 | 0.790 | 0.887 | 0.778 | 0.779 | 0.454 | 0.816 | 0.702 | 0.721 |
| | 25% | Merging | M-SMoE | 0.000 | 0.000 | 0.000 | 0.551 | 0.768 | 0.883 | 0.761 | 0.733 | 0.418 | 0.848 | 0.701 | 0.708 |
| | | | HC-SMoE | 0.788 | 0.269 | **0.529** | 0.470 | 0.713 | 0.833 | 0.622 | 0.646 | 0.376 | 0.805 | 0.665 | 0.641 |
| | | Pruning | Frequency | 0.000 | 0.000 | 0.000 | 0.548 | 0.789 | 0.889 | 0.775 | 0.735 | 0.438 | 0.801 | 0.694 | 0.709 |
| | | | EAN | 0.000 | 0.000 | 0.000 | 0.569 | 0.802 | 0.889 | 0.774 | 0.735 | 0.438 | 0.801 | 0.697 | **0.713** |
| | | | REAP | 0.763 | 0.253 | 0.508 | 0.555 | 0.771 | 0.864 | 0.740 | 0.736 | 0.452 | 0.805 | 0.693 | 0.702 |
| | 50% | Merging | M-SMoE | 0.000 | 0.000 | 0.000 | 0.262 | 0.348 | 0.693 | 0.479 | 0.237 | 0.290 | 0.523 | 0.542 | 0.422 |
| | | | HC-SMoE | 0.688 | 0.209 | **0.449** | 0.316 | 0.495 | 0.715 | 0.354 | 0.422 | 0.282 | 0.603 | 0.536 | 0.465 |
| | | Pruning | Frequency | 0.000 | 0.000 | 0.000 | 0.349 | 0.488 | 0.782 | 0.672 | 0.503 | 0.364 | 0.588 | 0.619 | 0.545 |
| | | | EAN | 0.000 | 0.000 | 0.000 | 0.480 | 0.736 | 0.876 | 0.760 | 0.607 | 0.424 | 0.762 | 0.694 | **0.667** |
| | | | REAP | 0.329 | 0.104 | 0.217 | 0.404 | 0.616 | 0.828 | 0.643 | 0.517 | 0.360 | 0.632 | 0.637 | 0.580 |
| Mixtral-8x7B-Instruct-v0.1 | **Baseline** | | | 0.469 | 0.123 | 0.296 | 0.650 | 0.842 | 0.887 | 0.861 | 0.691 | 0.496 | 0.722 | 0.740 | 0.736 |
| | 25% | Merging | M-SMoE | 0.296 | 0.044 | 0.170 | 0.532 | 0.775 | 0.828 | 0.746 | 0.529 | 0.424 | 0.603 | 0.632 | 0.634 |
| | | | HC-SMoE | 0.385 | 0.121 | **0.253** | 0.608 | 0.811 | 0.876 | 0.838 | 0.631 | 0.484 | 0.736 | 0.726 | **0.714** |
| | | Pruning | Frequency | 0.370 | 0.070 | 0.220 | 0.612 | 0.816 | 0.868 | 0.836 | 0.593 | 0.482 | 0.675 | 0.739 | 0.703 |
| | | | EAN | 0.360 | 0.092 | 0.226 | 0.613 | 0.814 | 0.875 | 0.842 | 0.613 | 0.498 | 0.690 | 0.733 | 0.710 |
| | | | REAP | 0.371 | 0.088 | 0.230 | 0.590 | 0.810 | 0.878 | 0.835 | 0.638 | 0.468 | 0.736 | 0.710 | 0.708 |
| | 50% | Merging | M-SMoE | 0.000 | 0.000 | 0.000 | 0.260 | 0.460 | 0.614 | 0.395 | 0.240 | 0.302 | 0.527 | 0.526 | 0.416 |
| | | | HC-SMoE | 0.152 | 0.033 | **0.093** | 0.540 | 0.764 | 0.862 | 0.795 | 0.544 | 0.448 | 0.675 | 0.709 | 0.667 |
| | | Pruning | Frequency | 0.156 | 0.008 | 0.082 | 0.504 | 0.739 | 0.793 | 0.771 | 0.463 | 0.426 | 0.675 | 0.646 | 0.627 |
| | | | EAN | 0.127 | 0.008 | 0.068 | 0.550 | 0.756 | 0.842 | 0.804 | 0.529 | 0.460 | 0.726 | 0.716 | 0.673 |
| | | | REAP | 0.121 | 0.022 | 0.071 | 0.531 | 0.779 | 0.869 | 0.787 | 0.543 | 0.460 | 0.773 | 0.697 | **0.680** |

Table A9: $\tau^2$-bench results with REAP compression across different benchmark domains on Qwen3-480B-A35B-Coder-FP8.

| Dataset | Compression | Method | pass^1 | pass^2 | pass^3 |
|---|---|---|---|---|---|
| Retail | **Baseline** | | 0.643 | 0.544 | 0.500 |
| | 25% | REAP | 0.661 | 0.535 | 0.465 |
| | 50% | REAP | 0.632 | 0.515 | 0.456 |
| Airline | **Baseline** | | 0.460 | 0.340 | 0.280 |
| | 25% | REAP | 0.487 | 0.367 | 0.320 |
| | 50% | REAP | 0.447 | 0.333 | 0.280 |
| Telecom | **Baseline** | | 0.500 | 0.398 | 0.325 |
| | 25% | REAP | 0.529 | 0.456 | 0.421 |
| | 50% | REAP | 0.471 | 0.339 | 0.263 |

Table A10: Additional $\tau^2$-bench results (pass^1 scores) across non-reasoning and reasoning models and compression levels with REAP. Interleaved thinking is not applied for MiniMax-M2 (thinking traces from previous turns are discarded in multi-turn trajectories).

| Model | Dataset | Baseline | 20% | 25% | 30% | 40% |
|---|---|---|---|---|---|---|
| Qwen3-Coder-30B-A3B | Airline | 0.393 | 0.407 | – | – | – |
| | Retail | 0.626 | 0.620 | – | – | – |
| | Telecom | 0.336 | 0.322 | – | – | – |
| GLM-4.5-Air | Airline (Thinking off) | 0.633 | – | 0.640 | – | – |
| | Retail (Thinking off) | 0.728 | – | 0.751 | – | – |
| | Telecom (Thinking off) | 0.284 | – | 0.307 | – | – |
| | Telecom (Thinking on) | 0.272 | – | 0.269 | – | – |
| MiniMax-M2 | Telecom | 0.591 | – | 0.576 | 0.591 | 0.553 |

Table A11: Additional BFCLv3 results across across non-reasoning and reasoning models and compression levels with REAP. Interleaved thinking is not applied for MiniMax-M2 (thinking traces from previous turns are discarded in multi-turn trajectories).

| Model | Benchmark | Baseline | 20% | 25% | 30% | 40% |
|---|---|---|---|---|---|---|
| Qwen3-Coder-30B-A3B | BFCLv3 | 0.632 | 0.622 | – | – | – |
| GLM-4.5-Air | BFCLv3 (Thinking) | 0.768 | – | 0.763 | – | – |
| GLM-4.6-FP8 | BFCLv3 (Thinking) | 0.784 | – | 0.773 | 0.768 | 0.742 |
| MiniMax-M2 | BFCLv3 | 0.626 | – | 0.615 | 0.599 | 0.579 |

Table A12: Kimi-Linear-48B-A3B-Instruct results at 30% REAP compression across coding, math, and long-context evaluations.

| Model | Compression | Coding | | | Math | | Long-Context | |
|---|---|---|---|---|---|---|---|---|
| | | HumanEval+ | MBPP+ | LiveCodeBench | MATH-500 | GSM8K | LongBench v2 | FRAMES |
| Kimi-Linear-48B-A3B-Instruct | Baseline | 0.823 | 0.669 | 0.276 | 0.818 | 0.873 | 0.368 | 0.557 |
| | 30% | 0.811 | 0.693 | 0.302 | 0.808 | 0.858 | 0.372 | 0.523 |

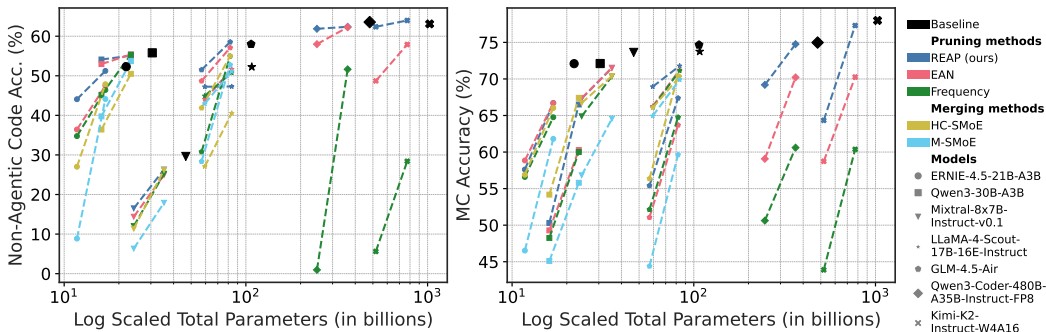

Figure A6: **Coding and MC accuracy across all models vs. parameters.** The benefits of REAP over other compression methods are evident at 50% compression. For large-scale SMoEs, REAP is near-lossless whereas the shortcomings of frequency-based pruning become apparent.

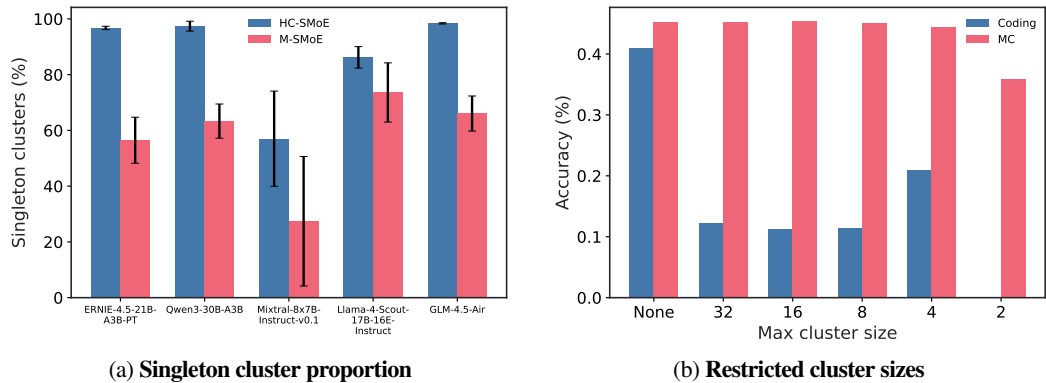

(a) **Singleton cluster proportion**  (b) **Restricted cluster sizes**

Figure A7: (a) **Average proportion of singleton clusters vs. model** for HC-SMoE and M-SMoE. We find that the clustering algorithms used by our baseline merging methods tend to generate a high proportion of singleton clusters containing just a single expert. In order to achieve the desired compression ratio, the large number of singletons conversely results in some clusters which contain many experts, in some cases $N/2 + 1$ experts for a layer with $N$ experts are grouped into a single cluster. (b) **Accuracy vs. maximum cluster size** using M-SMoE to compress 50% of experts in Qwen3-30B. While MC accuracy remains stable up to a maximum cluster size of 4, generative coding capabilities are severely diminished by restricting the clustering algorithm.

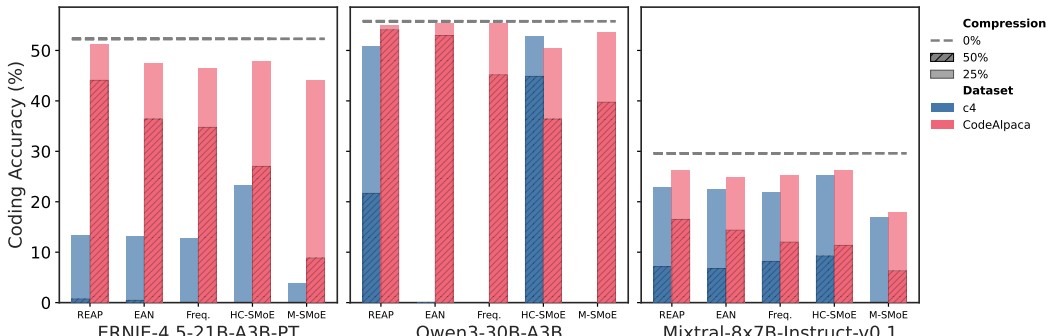

Figure A8: **Coding accuracy vs. calibration dataset**. Using domain-specific calibration datasets substantially improves compressed model quality within the target domain. Fine-grained models such as Qwen3-30B and ERNIE suffers greater degradation, with several compression methods failing to produce any coherent output when calibrated on C4.

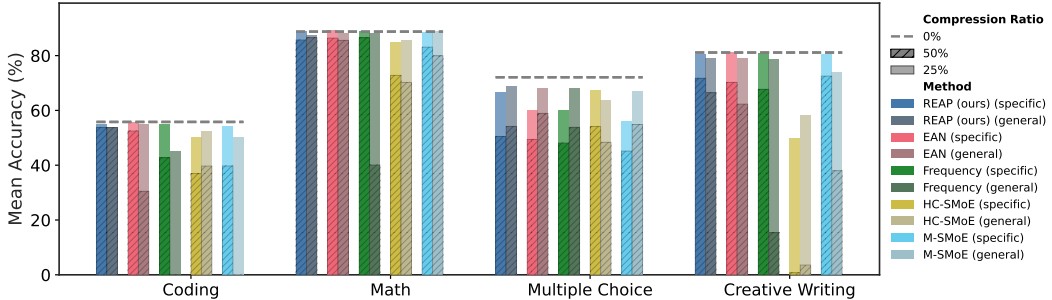

Figure A9: **Mean accuracy vs. task type for models calibrated with domain specific data versus general data.** The "general" calibration data consists of the combination of evol-codealpaca-v1, Writing-Prompts curated, and tulu-3-sft-personas-math and includes three times the total number of samples as the domain-specific calibration datasets. Compared to other compression methods, REAP best preserves accuracy across tasks when calibrated on the general dataset.

