# OpenReview forum: "REAP the Experts: Why Pruning Prevails for One-Shot MoE compression"
_ICLR.cc/2026/Conference — ICLR 2026 Poster_

### Official Review · Reviewer_Yea5 · 2025-10-29

**Soundness:** 3
**Presentation:** 3
**Contribution:** 3
**Rating:** 8
**Confidence:** 5

**Summary:**

This paper challenges the recent trend favoring expert merging for compressing Sparse MoE models, arguing instead that expert pruning is superior for generative tasks. The authors provide theoretical analysis showing that merging causes "functional subspace collapse" due to loss of independent router control over experts, while pruning preserves this control. They propose REAP (Router-weighted Expert Activation Pruning), a novel pruning criterion combining router gate values and expert activation norms. Experiments across models from 20B to 1T parameters demonstrate REAP's advantages on generative benchmarks, achieving near-lossless compression at 50% on code generation tasks.

**Strengths:**

* The theoretical insight in this work, that expert merging introduces irreducible error proportional to router policy variability, is insightful, novel, and well-presented.

* Evaluation is extensive across many models and tasks

* REAP is simple to implement and computationally efficient

* The paper is overall well-structured with good motivation and intuitive explanations

**Weaknesses:**

* Theorem 1's proof assumes "independence between the router policy and expert functions." This is barely possible in trained MoE models where routers and experts are jointly optimized.

* The proposed method does not always outperform expert merging baselines, such as Qwen3-30B-A3B on WildBench at 50% ratio.

* I would love to see more comparisons with other MoE-specific compression methods, such as quantization [1] [2].


[1] Li, P., Jin, X., Tan, Z., Cheng, Y. and Chen, T., 2024. QuantMoE-Bench: Examining Post-Training Quantization for Mixture-of-Experts. arXiv preprint arXiv:2406.08155.

[2] Duanmu, H., Li, X., Yuan, Z., Zheng, S., Duan, J., Zhang, X. and Lin, D., 2025. MxMoE: Mixed-precision Quantization for MoE with Accuracy and Performance Co-Design. arXiv preprint arXiv:2505.05799.

**Questions:**

See above

---

> ### Author Response · Authors · 2025-11-24
>
> We sincerely thank Reviewer Yea5 for their review. We are particularly pleased that you found our theoretical insights to be “insightful, novel, and well-presented”, our empirical evaluation to be “extensive across many models and tasks”, and that the paper was “well-structured with good motivation and intuitive explanations”. Below, we address your noted weaknesses:
>
> > Theorem 1's proof assumes "independence between the router policy and expert functions." This is barely possible in trained MoE models where routers and experts are jointly optimized.
>
> We agree that strict independence between router scale ($g_i + g_j$), router policy ($r(x)-\alpha$), and expert gap ($f_i(x) - f_j(x)$) is unlikely to hold perfectly in practice. This assumption was introduced to enable tractable analysis and isolate distinct error sources. We recognize it is a simplification. Crucially, our core insight remains valid even under correlation: merging is uniquely penalized by policy variability (Var[r(x)]), a term absent from pruning error. This fundamental difference persists regardless of the independence assumption's validity. In revision, we will reframe Section 3 as "mathematical intuition" developed under simplifying assumptions rather than as formal proofs. We will explicitly acknowledge that while independence enables clean decomposition, the key insight that merging loses router control while pruning preserves it is validated by our extensive empirical results (Section 5) in realistic settings where strict independence does not hold.
>
>
> > The proposed method does not always outperform expert merging baselines, such as Qwen3-30B-A3B on WildBench at 50% ratio.
>
> You are correct that M-SMoE outperforms REAP for Qwen3-30B-A3B creative writing at 50% compression and we acknowledge there are other specific instances of model/task combinations where one of the other compression methods outperforms REAP. However, REAP's key advantage is its consistency across diverse settings. Examining Tables 2 and 3: at 50% compression, REAP achieves rank-1 or rank-2 performance in 26 of 28 model/compression/task combinations, whereas HC-SMoE and M-SMoE show high variance, performing well on some configurations but catastrophically failing on others (e.g., M-SMoE on GLM-4.5-Air coding drops to 29.6% vs. REAP's 55.3%). REAP offers reliable results across models without risking catastrophic failure.
>
> > I would love to see more comparisons with other MoE-specific compression methods, such as quantization [1] [2].
>
> Your question regarding how REAP performs compared to other expert compression baselines such as quantization is well motivated. Crucially, we argue that expert pruning is composable with quantization and demonstrate this empirically with our large-scale expert pruning evaluations where we calibrate, prune, and evaluate Qwen3-Coder-480B and Kimi-K2 with FP8 and W4A16 quantization, respectively. This is particularly relevant in comparison to expert merging as in that setting the fine-grained shared quantization scales across groups of parameters must be reconciled during merging whereas pruning requires no such reconciliation.
>
> Generally, we expect quantization to offer superior accuracy at a given compression ratio up to 4-bit weights. For instance, after quantizing Qwen3-30B-A3B and Mixtral-8x7B from BF16 to W4A16-G128 using AWQ, we obtain EvalPlus accuracies of 84.3% and 47.82%, respectively, compared to 82.1% and 23.5% for REAP at 50% compression. The 4-bit weight quantization represents a ~75% compression rate, a rate at which expert pruning alone is unable to maintain reasonable accuracy.
>
> However, combined together expert pruning and quantization enable compression rates beyond which either method in isolation can obtain. For example, our Kimi-K2 results include both 4-bit weights and up to 50% expert compression. This represents a total compression factor of 87.5% which would require quantization to 2-bits to directly compare with, which are not natively supported on current accelerators.
>
> Even if such support were available for 2-bit weights, preliminary results suggest that **REAP combined with 4-bit weights outperforms more aggressive quantization**. Using llm-compressor, we applied AWQ to quantize Qwen3-30B-A3B to W2A16-G128 and compared it to W4A16-G128 with 50% of experts pruned via REAP and evaluated on EvalPlus. The following table summarizes our results:
> | Model                         |   Eval+ coding accuracy |
> |:------------------------------|------------------------:|
> | Qwen3-30B-A3B-W2A16           |                   0.304 |
> | Qwen3-30B-A3B-W4A16-50% REAP  |                   0.820  |
> | Qwen3-30B-A3B-W16A16-50% REAP |                   0.821 |
>
>
> We acknowledge that a more thorough comparison with different low-bit schemes, quantization algorithms, and expert pruning combinations is an exciting and interesting topic for future work.

---

### Official Review · Reviewer_5xpD · 2025-10-30

**Soundness:** 2
**Presentation:** 2
**Contribution:** 3
**Rating:** 2
**Confidence:** 4

**Summary:**

This paper proposes REAP (Router-weighted Expert Activation Pruning) for pruning experts in Mixture-of-Experts model. They show thorough empirical validation that pruning outperforms merging methods on generative tasks. REAP prunes MoE experts based on router gates and activation norms. The central claim seems to be that merging causes "functional subspace collapse" by losing the router's input-dependent control, while pruning preserves independent expert modulation.

**Strengths:**

1) Well written introduction (and comprehensive related work) of concepts like model merging, expert pruning etc.
2) The intuition is sound in section 3.1: merging forces a static convex combination when you need a dynamic one.
3) I appreciate the effort to formalize the merging and pruning problem and provide mathematical intuition for why pruning outperforms merging.
3) Strong/thorough empirical validation.

**Weaknesses:**

In general, I understand the authors' core ideas behind REAP, and empirical results are convincing. My major concern is that the mathematical presentation has significant issues (in my view). Perhaps there needs to major rewording where authors state them as "mathematical intuition" rather than formal proofs, explicitly acknowledging the simplifying assumptions while emphasizing that the approach is validated by strong empirical evidence.

Major Weaknesses
1) I am not convinced by Section 3.3 experiments and the related claim. Authors show PCA (PC1, PC2) plots to claim it as the empirical evidence for loss of independent control of "degrees of freedom" when one does model merging vs pruning.
- PCA by definition is reducing from $n$ dimensions to $k << n$ dimension ($k=2$) and what if independent control is preserved in dimensions 3-100 but lost in the first 2 PCs? What about variance? (should report variance explained by PC1+PC2)
- Since this is towards understanding functional subspace (potentially high dimensional) a more convincing setup/experiments would require quantitative metrics like: rank of expert output matrix, any information-theoretic metric etc.
- Current setup would be stronger as "preliminary empirical observations" rather than conclusive evidence.
- The real evidence should come from performance metrics and more direct measures of expert independence.

2) Missing Rigor in "Irreducible Error" Claim in general: I have similar issues in this part as well as point 3 below. I understand the intuitive sketch but I think stating it as formal proof is a bit of stretch. For this to be a rigorous claim one needs to proof that 1) that this is a lower bound over all possible merged representations or 2) this is the optimal form or that no other parameterization could do better.

3) Theorem 1: Despite stating the independence assumption I believe there are mathematical errors in this and subsequent sections. For example: authors write the $||\Delta_{i,j}||^2$ outside the expectation, but $\Delta_{i,j}$ depends on $x$, so that expression does not look correct (at-least without more explanation). Similarly the three terms in Equation 5 are not rigorous and one cannot generally factor this into three separate expectations without making strong independence assumptions (which are also not stated completely) . Equation 6 also seems to have similar issues.

4) *However, since our goal is to prune unimportant experts, we can reasonably assume their gate-values are small w* --> At $50\\%$ compression, one is pruning half the experts - Some of these will NOT have small gate values - authors use the approximation to derive the metric, then apply it globally - so I believe the explanation in this section is not entirely correctly stated with missed statements.

5) There is a disconnect between Section 3 and 4. Whatever the authors use in section 3 does not reflect in Section 4 (ex: no mention/use of expert differences, policy variability? among other issues.)

6) Not sure if I missed this but it's hard to see in Figure 2 the claim of near-lossless compression at any task. Ex: wrt to EAN (another method). I saw that the difference between the two is not much? Since this is an impactful claim - it might be better or more cleanly demonstrated?

Minor Weaknesses
1) Section 3.1: the equations or the explanation in general in this section can be cleared a bit (especially *"define the router’s input-dependent mixing ratio ..."*).

**Questions:**

1) Section 5 (experimental setup):  *We compress models by pruning or merging 25% or 50% of experts in each layer* - are these experts selected at random (for non M-SMoEs)?

---

> ### Author Response · Authors · 2025-11-24
>
> Thank you for your highly insightful and constructive review. We are greatly encouraged that you recognize the fundamental challenge we address and the success of our proposed method. We are particularly pleased that you highlighted the following core strengths of our submission: the clarity and context provided by our "Well written introduction (and comprehensive related work)" , the "sound intuition in section 3.1" regarding static vs. dynamic convex combinations, the appreciation for our "effort to formalize the merging and pruning problem and provide mathematical intuition" , and that our work features "Strong/thorough empirical validation" and that our "empirical results are convincing". Your major concern, that "the mathematical presentation has significant issues" , is extremely valuable, and we fully agree with your recommendation to revise the language from "formal proofs" to "mathematical intuition". We have carefully considered all your questions and concerns, providing detailed responses below:

---

> > ### Author Response · Authors · 2025-11-24
> >
> > > 1. "I am not convinced by Section 3.3 experiments... Authors show PCA (PC1, PC2) plots to claim it as the empirical evidence for loss of independent control... what if independent control is preserved in dimensions 3-100... What about variance? (should report variance explained by PC1+PC2)... a more convincing setup/experiments would require quantitative metrics like: rank of expert output matrix, any information-theoretic metric etc."
> >
> > We agree with the reviewer that the PCA plots in Section 3.3 are qualitative observations rather than definitive proof, and we will revise the text to state this explicitly. PCA necessarily compresses a high dimensional functional subspace into a low dimensional projection, so our plots cannot rule out independent control in higher components; we will add this limitation to the caption and main text. We will add a table of cumulative variance explained for PC1 and PC2 to Appendix A4. To facilitate our rebuttal discussion in the interim, we have included the table below which summarizes the cumulative total variance explained of PC1 and PC2:
> >
> > | Model         |   Layer |   Baseline |   Merged |   Pruned |
> > |:--------------|--------:|-----------:|---------:|---------:|
> > | Qwen3-30B-A3B |       0 |     0.2343 |   0.27   |   0.1845 |
> > | Qwen3-30B-A3B |      47 |     0.7195 |   0.7437 |   0.686  |
> > | ERNIE-4.5-21B |       0 |     0.3836 |   0.2851 |   0.2733 |
> > | ERNIE-4.5-21B |      26 |     0.2563 |   0.4599 |   0.0785 |
> > | Llama-4-Scout |       0 |     0.9032 |   0.9343 |   0.848  |
> > | Llama-4-Scout |      47 |     0.9473 |   0.9546 |   0.8754 |
> > | Mixtral-8x7B  |       0 |     0.6486 |   0.8479 |   0.4016 |
> > | Mixtral-8x7B  |      31 |     0.858  |   0.814  |   0.7027 |
> >
> > From the above, we highlight the following: 1) For coarse-grained SMoEs such as Llama-4-Scout and Mixtral with few experts per layer, PCA1 and PC2 capture most of the variance in the activations. Even in fine-grained SMoEs with many experts per layer such as Qwen3 and ERNIE, a large portion of the total variance is captured by PC1 and PC2; 2) The merged variance explained is consistently higher than the baseline, suggesting that the merged outputs have lost some of their high-dimensional complexity; and 3) The pruned variance explained is consistently lower than baseline, suggesting that pruning preserves outlier experts and the high-dimensional complexity of the baseline model.
> >
> > Additionally, we now provide additional quantitative evidence that expert merging causes greater distortion to the original expert output manifold compared to pruning. We project the expert activations onto the unit hypersphere and measure the Wasserstein distance between the original and compressed expert output manifolds. Across all models, the merged outputs are further from the original manifold demonstrating that merging fundamentally alters the output manifold to a greater degree than pruning. See the table below for the average Wasserstein distance across layers for each model. We will additionally include plots of the Wasserstein distance vs. layer index for each model in the revised paper.
> >
> > | Model         |   Merged Wasserstein Dist. |   Pruned Wasserstein Dist. |
> > |:--------------|---------------------------:|---------------------------:|
> > | Qwen3-30B-A3B |                     0.4512 |                     0.3183 |
> > | ERNIE-4.5-21B |                     0.4285 |                     0.2827 |
> > | Llama-4-Scout |                     0.1711 |                     0.0961 |
> > | Mixtral-8x7B  |                     0.3178 |                     0.1583 |
> >
> > We will also clarify that our functional output analysis is not our only primary evidence for loss of independent control. Instead, we will point readers to Figures 3a, where merged models exhibit significantly lower N-gram diversity. Here we also include a new analysis in a newly added Figure 3c that demonstrates that the merged model output logits diverge more rapidly from the baseline (higher JSD) than under pruning as a function of completion token position.

---

> > > ### Author Response · Authors · 2025-11-24
> > >
> > > > 2. "Missing Rigor in 'Irreducible Error' Claim in general: I understand the intuitive sketch but I think stating it as formal proof is a bit of stretch. For this to be a rigorous claim one needs to proof that 1) that this is a lower bound... or 2) this is the optimal form..."
> > >
> > > We appreciate the reviewer’s careful reading of our “irreducible error” discussion. We agree that, in its current phrasing, the section can be interpreted as making a fully general lower-bound claim, which is stronger than what our assumptions justify. Our goal is to provide a formalized intuition for why merging collapses certain router-dependent degrees of freedom that pruning preserves, under simplifying assumptions. Concretely, we revise this part of the paper to (i) explicitly label the result as “mathematical intuition under simplifying assumptions” rather than as a general theorem, (ii) list the precise assumptions that are used in the derivation (e.g., the independence structure needed for the factorizations in Eqs. (5)-(6) and the restricted functional class of merged experts we consider), and (iii) restate the conclusion as a conditional statement that holds under these assumptions, avoiding language such as “irreducible error of any merged model” that would suggest a universal lower bound over all possible merged parameterizations. In the camera-ready version we will make these revisions explicit in Section 3, so that the scope of the argument is clearly presented as a simplified, assumption-driven intuition rather than a fully general optimality theorem.

---

> > > > ### Author Response · Authors · 2025-11-24
> > > >
> > > > > 3. Theorem 1: Despite stating the independence assumption I believe there are mathematical errors in this and subsequent sections. For example: authors write the outside the expectation, but depends on , so that expression does not look correct (at-least without more explanation). Similarly the three terms in Equation 5 are not rigorous and one cannot generally factor this into three separate expectations without making strong independence assumptions (which are also not stated completely) . Equation 6 also seems to have similar issues.
> > > >
> > > > We thank the reviewer for flagging the issues around Theorem 1 and the associated expectations and independence assumptions. We have revisited these derivations and agree that the dependencies and assumptions should be stated more carefully. For the merging error, we will write the full expectation correctly as:
> > > >
> > > > $$\mathcal{E}\_{\text{merge}} = \mathbb{E}\_{x}\left[(g_i(x) + g_j(x))^2 \cdot (r(x) - \alpha^\star)^2 \cdot \|\Delta_{ij}(x)\|_2^2\right]$$
> > > >
> > > > where $\alpha^\star = \mathbb{E}[r(x)]$.
> > > > To obtain the three-term decomposition you correctly identified as requiring justification (Eq. 5), we now explicitly state our factorization assumption: "We assume that router scale $(g_i(x) + g_j(x))^2$, policy variability $(r(x) - \alpha^\star)^2$, and expert differences $\|\Delta_{ij}(x)\|\_2^2$ are approximately uncorrelated across inputs."
> > > > Only under this assumption can we decompose the error as:
> > > >
> > > > $$\mathcal{E}\_{\text{merge}} \approx \underbrace{\mathbb{E}\_x[(g_i(x)+g_j(x))^2]}\_{\text{router scale}} \cdot \underbrace{\mathrm{Var}[r(x)]}\_{\text{policy variability}} \cdot \underbrace{\mathbb{E}\_x[\|\Delta_{ij}(x)\|_2^2]}\_{\text{expert gap}}$$
> > > >
> > > > This decomposition reveals that the error of merging is proportional to: 1) how frequently the router uses these experts; 2) how dynamically the router mixes between them; and 3) how functionally different they are. These three factors correspond exactly to our original Eq. 5, but we will now present them correctly as a decomposition under stated assumptions rather than as separate expectations.
> > > >
> > > > For the pruning error (Eq. 6), we have refined our analysis to include the error contribution of *promoting* a previously inactive expert to the top-$K$ experts whenever the pruned expert was originally in the top-$K$ set. This decomposition reveals that the overall error due to pruning expert $j$ and promoting expert $i$ into the top-$K$ set is
> > > >
> > > > $$ \mathcal{E}\_{prune} = \mathbb{E}\_{x| j \in \mathcal{T} (x)}\big[\big\|\underbrace{g_j(x) f_j(x) - g_i(x) f_i(x)}\_{\text{substitution  error}} -  \underbrace{\frac{g_j(x) - g_i(x)}{1-g_j(x) + g_i(x)} \sum_{k \neq i,j} g_k(x) f_k(x)}\_{\text{renormalization error}}  \big\|^2_2\big]$$
> > > >
> > > > where $\mathcal{T}(x) = TopK(g(x))$ and $g_i(x)$ in this expression refers to expert $i$'s gate-value after promotion to the top-$K$ set (previously zeroed when it was at position K+1). When expert $j$ is at the top-$K$ boundary, these gate-values satisfy $g_j(x) \approx g_i(x)$ which directly minimizes the renormalization error. This refined derivation highlights that merging is uniquely penalized by the router policy ($\mathrm{Var}[r(x)]$) and pruning by the magnitude of the substitution between the pruned and promoted expert outputs. By the triangle inequality, the magnitude of the substitution error vector is upper bounded by
> > > >
> > > > $$||g_j(x)f_j(x) - g_i(x)f_i(x)|| \leq g_j(x)||f_j(x)|| + g_i(x)||f_i(x)||.$$
> > > >
> > > > With top-$k$ routing $g_i \leq g_j$ and the worst case error occurs when $g_j = g_i$
> > > >
> > > > $$|| g_j(x) f_j(x) - g_i(x) f_i(x)|| \leq  g_j(x) \big( ||f_j(x)|| + || f_i(x) ||\big).$$
> > > >
> > > > Since the identity of the promoted expert $i$ and thus $||f_i(x)||$ varies across tokens, directly minimizing the pruned expert's impact $g_j ||f_j(x)||$ is an effective heuristic to minimize the upper bound of the total error.
> > > >
> > > > This analysis is supported by our empirical findings. Highly-granular SMoEs with many experts per layer (Qwen3, ERNIE) use highly variable routing polices (high $\mathrm{Var}[r(x)]$) to combine many small contributions (small $g_j$). This is precisely the setting in which we expect merging to incur a high error and pruning a lower error. This is supported by our observations in Section 5 which show that the relative performance gains of REAP over expert merging are maximized on high-granular SMoEs (Qwen3, ERNIE, Kimi-K2) and minimized on low-granularity SMoEs (Llama-4 and Mixtral).

---

> > > > > ### Author Response · Authors · 2025-11-24
> > > > >
> > > > > *Continued from previous comment:*
> > > > >
> > > > > Following your suggestion, in the revised text we will reframe our analysis as a motivation under simplifying assumptions rather than a formal proof. We will present all errors as full expectations before any factorization, explicitly state the independence assumptions required, and clarify that this analysis provides intuition validated by extensive empirical results. The strong performance gap between pruning and merging across 7 models (Tables 2-3), particularly the 97\%+ retention for REAP versus <90\% for merging at 50\% compression, demonstrates that these mathematical insights, despite their simplified derivation, capture essential differences between the two operations.

---

> > > > > > ### Author Response · Authors · 2025-11-24
> > > > > >
> > > > > > > 4. However, since our goal is to prune unimportant experts, we can reasonably assume their gate-values are small w --> At  compression, one is pruning half the experts - Some of these will NOT have small gate values - authors use the approximation to derive the metric, then apply it globally - so I believe the explanation in this section is not entirely correctly stated with missed statements.
> > > > > > > 5. There is a disconnect between Section 3 and 4. Whatever the authors use in section 3 does not reflect in Section 4 (ex: no mention/use of expert differences, policy variability? among other issues.)
> > > > > >
> > > > > > We appreciate this important clarification. The reviewer is correct that at 50\% compression, not all pruned experts will have negligible gate-values. The small gate-value condition ($\mathbb{E}_x[g_j(x)] \ll 1$) was a simplifying assumption used only to motivate a tractable saliency metric from the complex, non-linear pruning error (Eq. 7). However, our refined analysis of the expected error of pruning provides a more rigorous justification for the REAP metric that holds even when gate-values are not negligible and explicitly connects Sections 3 and 4.
> > > > > >
> > > > > > As derived in reponse to Weakness 3, the substitution error is bounded by $g_j ( ||f_j|| + || f_i||)$. This inequality reveals that the error is limited by the pruned gate magnitude $g_j$ scaling the sum of the pruned and promoted expert norms. This confirms REAP explicitly minimizes the known components of the error bound ($g_j\||f_j\||$) while simultaneously shrinking the coefficient ($g_j$) that scales the unknown component ($\||f_i\||$). REAP effectively minimizes the worst-case substitution error regardless of the absolute compression ratio.
> > > > > >
> > > > > > In our revised paper, we will motivate REAP in based on our updated analysis and remove any assumptions regarding the magnitude of $g_j$. To strengthen the manuscript's coherence, we will add a bridging paragraph at the end of Section 3 that explicitly maps each theoretical concept (policy variability vs. substitution magnitude) to the components of the REAP criterion. We will also include a remark explaining that the superiority of REAP over EAN (which only uses activations) and frequency-based pruning (which only uses gates) empirically validates our theoretical finding that both components matter.
> > > > > >
> > > > > > > 6. Not sure if I missed this but it's hard to see in Figure 2 the claim of near-lossless compression at any task. Ex: wrt to EAN (another method). I saw that the difference between the two is not much? Since this is an impactful claim - it might be better or more cleanly demonstrated?
> > > > > >
> > > > > > We thank the reviewer for asking for a clearer demonstration of the “near-lossless” claim. We agree that differentiating between some compression methods in Figure 2 is challenging due to the shared y-axis across disparate tasks. These results are also tabulated in Table 2 which facilitates a more clear comparison, we will add a comment in the caption of Figure 2 which refers to Table 2 in the camera-ready version.
> > > > > >
> > > > > > Our main claims of “near-lossless” accuracy rests on the large-scale models (Qwen3-Coder-480B and Kimi-K2-Instruct) shown in Table 3 as summarized below:
> > > > > > * Qwen3-Coder-480B (Non-Agentic Code Avg):
> > > > > >   * Baseline: 0.660
> > > > > >   * REAP (ours): 0.644 (loss of 1.6 points)
> > > > > >   * EAN: 0.607 (loss of 5.3 points)
> > > > > >   * Frequency: 0.011 (catastrophic failure)
> > > > > > * Kimi-K2-Instruct-W4A16 (Non-Agentic Code Avg):
> > > > > >   * Baseline: 0.659
> > > > > >   * REAP (ours): 0.646 (loss of 1.3 points)
> > > > > >   * EAN: 0.513 (loss of 14.6 points)
> > > > > >   * Frequency: 0.062 (catastrophic failure)
> > > > > >
> > > > > > The term “near-lossless” is justified here because REAP retains over 97% of the baseline model's model quality on these critical tasks after removing 50% of the parameters, while EAN suffers a loss of $\gt 8$\% and frequency-based pruning fails entirely. In the revised version, we will: 1) ensure the figures and text clearly link the “near-lossless” claim to the results in Table 3 and; 2) tighten the wording in the discussion to explicitly state the model quality performance margins (e.g., “loss of $\leq 2$\%”) to make the claim precise and easily verifiable.

---

> > > > > > > ### Author Response · Authors · 2025-11-24
> > > > > > >
> > > > > > > > 7. Section 3.1: the equations or the explanation in general in this section can be cleared a bit (especially "define the router’s input-dependent mixing ratio ...").
> > > > > > >
> > > > > > > We thank the reviewer for pointing out the ambiguity in Section 3.1 regarding the mixing ratio. We agree that the definitions here can be made more precise and explicit. Our main intent in introducing $r(x)$ is to relate the dynamic mixing of experts $i$ and $j$ in the uncompressed model to a static, convex combination in the merged model. In our revision, we will standardize notation and explicitly clarify:
> > > > > > >
> > > > > > > * Top-K context - The layer output (Eq. 1) is a convex combination over the set of active experts, $\mathcal{T}(x)$.
> > > > > > > * Binary Ratio $r(x)$ - We will clearly state that the input-dependent mixing ratio $r(x) := \frac{g_i(x)}{g_i(x)+g_j(x)}$ is defined locally over the two experts being analyzed ($i$ and $j$), and only applies on the support where both $i$ and $j$ are active. This isolates the dynamic mixing that merging fails to capture.
> > > > > > >
> > > > > > > These changes are purely expository and will ensure the conceptual link from the general MoE output to the specific pairwise analysis is transparent.
> > > > > > >
> > > > > > > > 8. Section 5 (experimental setup): We compress models by pruning or merging 25% or 50% of experts in each layer - are these experts selected at random (for non M-SMoEs)?
> > > > > > >
> > > > > > > We apologize for the lack of clarity. For non M-SMoE baselines, experts are not selected at random. In all pruning methods (including REAP, EAN , and frequency-based pruning ), we prune the experts with the smallest saliency scores according to the respective criterion. For merging methods like HC-SMoE, we follow the published procedures to cluster and merge experts based on distances and usage statistics. See Appendix D for formal definitions of each compression method studied and how the experts are selected for pruning / merging.

---

### Official Review · Reviewer_KnfQ · 2025-11-03

**Soundness:** 3
**Presentation:** 3
**Contribution:** 3
**Rating:** 6
**Confidence:** 2

**Summary:**

This paper investigates compression techniques for Sparse Mixture-of-Experts (SMoE) models, challenging recent claims that expert merging outperforms pruning. The authors provide theoretical analysis showing that merging introduces irreducible error through "functional subspace collapse" - essentially, when you merge experts, the router loses its ability to independently control them based on input. They propose REAP (Router-weighted Expert Activation Pruning), which selects experts to prune by considering both router gate values and activation norms. The method is evaluated across models ranging from 20B to 1T parameters, with testing on generative tasks like code generation, creative writing, and math reasoning. Results show REAP consistently outperforms merging methods on generative benchmarks, particularly at 50% compression ratios.

**Strengths:**

1. The authors offer a clean, intuitive analysis showing that merging induces an irreducible error proportional to router policy variability.

2. REAP’s design combines router gate strength with expert activation norms. It is conceptually simple yet empirically robust. It scales efficiently to trillion-parameter models and operates without fine-tuning.

3. The experiments span multiple model families, from 20B to 1T parameters, and cover diverse benchmarks (code, creative writing, math, tool use). The generative evaluation is a standout—showing that REAP maintains performance where merging sharply degrades.

4. The PCA visualizations in Figures 1 and A4 are interesting. They effectively demonstrate the functional collapse phenomenon and show how pruning maintains the geometric structure of the expert manifold while merging contracts everything toward the center.

**Weaknesses:**

1. This paper focuses on one-shot compression, which is fair for the “no retraining” setting, but most real-world deployments fine-tune after merging. It’s unclear whether the theoretical and empirical gaps between merging and pruning persist once merged experts are lightly retrained.

2. The independence assumption between router policy and expert function in the irreducible error derivation (Eq. 5) is strong and may not hold in practice, especially in late layers where experts co-adapt. This limits how far the theorem generalizes.

3. Section 5.1 highlights that calibration data domain critically affects compression success. This introduces a sensitivity that could complicate real deployment, but the paper doesn’t analyze how REAP behaves under mismatched calibration domains.

**Questions:**

1. Would merging still underperform pruning if merged experts were allowed limited post-merge fine-tuning? How much of the “functional subspace collapse” could be recovered through optimization?

2. How sensitive is REAP to the assumed linearity between gate-values and activation magnitudes? Would a nonlinear scaling or normalization improve robustness across layers?

3. Can REAP and merging be hybridized—e.g., pruning weak experts first and merging only within strongly correlated clusters—to achieve better compression-performance trade-offs?

---

> ### Author Response · Authors · 2025-11-24
>
> We thank Reviewer KNFQ for their positive feedback and constructive suggestions. We are encouraged that you found our analysis “clean, intuitive” and that our generative evaluation was a “standout” which demonstrated that “REAP maintains performance where merging sharply degrades”. Below, we address your noted weaknesses and questions:
>
> > This paper focuses on one-shot compression, which is fair for the “no retraining” setting, but most real-world deployments fine-tune after merging. It’s unclear whether the theoretical and empirical gaps between merging and pruning persist once merged experts are lightly retrained…Would merging still underperform pruning if merged experts were allowed limited post-merge fine-tuning? How much of the “functional subspace collapse” could be recovered through optimization?
>
> We agree that this is an important question. Our work focuses on the one-shot setting, which is valuable for local deployment, academic research, and resource-constrained environments where fine-tuning is too computationally demanding. Nevertheless, our preliminary findings outlined below demonstrate that **REAP one-shot pruning outperforms merged models after supervised fine-tuning (SFT)**. We fine-tuned our 50% compressed Qwen3-30B-A3B M-SMoE checkpoint calibrated on coding data. The SFT model was evaluated on non-agentic code generation tasks (EvalPlus and LiveCodeBench) and the following table compares the results to the one-shot merged and REAP pruned models:
>
> | Model           |   Non-agentic coding accuracy |
> |:----------------|------------------------------:|
> | One-shot merged |                         0.413 |
> | SFT merged      |                         0.458 |
> | One-shot pruned |                         0.557 |
>
> Despite fine-tuning, the merged model still underperforms one-shot REAP by nearly 10 percentage points. This persistent gap suggests that the functional subspace collapse from merging creates fundamental limitations that optimization cannot fully overcome; the lost degrees of freedom from tying router gates cannot be easily recovered through weight updates alone.
>
> The M-SMoE model was fine-tuned using a learning rate of 5e-5, cosine learning rate scheduler with 0.03 warm-up ratio, mixed precision, batch size of 4, and a maximum sequence length of 2048 tokens. We trained the model for 1 epoch on the entire CodeAlpaca dataset, only calculating loss on completion tokens. While further hyperparameter optimization or knowledge distillation may improve the fine-tuned merged model, the substantial gap (9.9%) indicates that the theoretical advantages of pruning persist in practice. The router's lost independent control over merged experts appears to be a fundamental architectural constraint rather than an optimization issue. We leave comprehensive fine-tuning comparisons for future work, but these initial results support our theoretical predictions.
>
> > The independence assumption between router policy and expert function in the irreducible error derivation (Eq. 5) is strong and may not hold in practice, especially in late layers where experts co-adapt. This limits how far the theorem generalizes.
>
> We agree that strict independence between router scale (g_i + g_j), router policy (r(x)-α), and expert gap (f_i(x) - f_j(x)) is unlikely to hold perfectly in practice. This assumption was introduced to enable tractable analysis and isolate distinct error sources. We recognize it is a simplification. Crucially, our core insight remains valid even under correlation: merging is uniquely penalized by policy variability (Var[r(x)]), a term absent from pruning error. This fundamental difference persists regardless of the independence assumption's validity. In revision, we will reframe Section 3 as "mathematical intuition" developed under simplifying assumptions rather than as formal proofs. We will explicitly acknowledge that while independence enables clean decomposition, the key insight that merging loses router control while pruning preserves it is validated by our extensive empirical results (Section 5) in realistic settings where strict independence does not hold.

---

> > ### Author Response · Authors · 2025-11-24
> >
> > > Section 5.1 highlights that calibration data domain critically affects compression success. This introduces a sensitivity that could complicate real deployment, but the paper doesn’t analyze how REAP behaves under mismatched calibration domains.
> >
> > You raise an important practical concern. All expert compression methods exhibit sensitivity to calibration data, though REAP shows greater robustness than alternatives. Figure A7 and Table A6 reveal model-specific patterns in domain sensitivity. For Qwen3-30B, REAP shows notable robustness at 25% compression, maintaining ~85% of its in-domain performance when using out-of-domain c4 data (48% vs 57% accuracy). However, other architectures like ERNIE-4.5 show severe degradation with domain mismatch, dropping from 44% to 14% accuracy. This suggests that domain sensitivity is influenced by both the compression method and the underlying model architecture.
> >
> > Figure A8 further explores this by comparing domain-specific versus "general" calibration (concatenating all domain datasets). At 25% compression, REAP maintains 95% of its domain-specific performance with general calibration, while M-SMoE drops to 71% and frequency-based pruning to 68%. However, at 50% compression, all methods benefit significantly from domain-specific calibration.
> >
> > For deployment, we recommend: 1) At 25% compression, REAP with general calibration provides robust performance across domains; 2) At 50% compression, domain-specific calibration is essential; and 3) When the deployment domain is unknown, conservative compression (25%) with diverse calibration data minimizes risk. Future work could explore automatic calibration domain selection or adaptive compression rates based on detected domain shifts.
> >
> >
> > > How sensitive is REAP to the assumed linearity between gate-values and activation magnitudes? Would a nonlinear scaling or normalization improve robustness across layers?
> >
> >
> > This is an insightful question that highlights an interesting design choice. The linear formulation in REAP ($g_k(x) \cdot \||f_k(x)\||$) has both theoretical and practical justification.
> >
> > Theoretically, this product directly measures each expert’s contribution to the layer output magnitude, as shown in our error analysis (Section 3). The linearity emerges naturally from the MoE formulation where the layer output is $\sum_k g_k(x) f_k(x)$. For architectures using normalized top-K routers (where $\sum_{k \in \text{top-k}} g_k(x) = 1$), we maintain consistency by using the same normalization in REAP.
> >
> > Empirically, this simple formulation proves remarkably robust across all tested architectures (from 20B to 1T parameters) without layer-specific tuning. However, your suggestion of nonlinear scaling is intriguing, particularly for addressing potential scale variations across layers. For instance, log-scaling might better handle the exponential growth in activation magnitudes sometimes observed in deeper layers, while square-root scaling could emphasize diversity over magnitude.
> >
> > While we have not systematically explored these variants given REAP’s strong model quality performance with linear formulation, we agree this represents promising future work. The optimal scaling might be architecture-specific or learnable. We appreciate this suggestion and will acknowledge it as a valuable direction for enhancing REAP’s robustness.
> >
> > > Can REAP and merging be hybridized—e.g., pruning weak experts first and merging only within strongly correlated clusters—to achieve better compression-performance trade-offs?
> >
> > Hybrid approaches are conceptually feasible and potentially promising. Our analysis in Figure A6a demonstrates that HC-SMoE already produces many singleton clusters (experts left unmerged) alongside "mega-clusters" containing dozens of experts. A principled hybrid might prune low-saliency experts, merge only within small, tightly-correlated clusters, and preserve high-saliency experts as singletons. This approach would avoid the functional collapse from merging diverse experts while still achieving memory savings from consolidating truly redundant ones. The key insight from our theory is that merging works when Var[r(x)] is low (experts combined in similar proportions across inputs) and $||\Delta_{ij}||$ is small (experts are functionally similar). While we have not implemented this hybrid, our results suggest it could achieve better compression-performance trade-offs than pure merging, particularly at moderate compression ratios (30-40%). This represents valuable future work, especially if combined with limited fine-tuning to restore router-expert coordination for the merged clusters.

---

### Author Response · Authors · 2025-11-26
**Thank you and revised paper**

We sincerely thank all reviewers for their time and efforts; your comments, questions, and suggestions have materially improved the work. We are greatly encouraged that reviewers found our motivation provided “clean, intuitive analysis” (KNFQ), “sound” intuition (5xpd), and was “insightful, novel, and well-presented” (Yea5). Our empirical results were universally praised by our reviewers as a “standout” (KNFQ), “strong/thorough” (5xpd), and “extensive across many models and tasks” (Yea5). We have responded to each reviewer's itemized weaknesses and questions below and encourage our reviewers to confirm whether their concerns have been adequately addressed.

Additionally, we have uploaded a revised copy of the paper which includes the following revisions:
* Reframed Section 3 and 4 as motivation rather than formal proofs, our intent is to build the reader's intuition regarding the sources of error which are present under expert merging and pruning. We have also explicitly highlighted any simplifying assumptions made in our analysis. All conclusions based on an analysis of our simplified model are now explicitly stated as being conditional on the stated assumptions.
* Improved the rigor of Section 3 by explicitly stating the expected errors for both merging and pruning in terms of the layerwise reconstruction error. This revision now enables a more holistic analysis of the terms that contribute to pruning error. We now state an upper bound on this error in terms of the pruned expert gate-value and the expert output norms of the pruned and “promoted” expert which we believe to be a more complete description of the pruning operation.
* We added Figure 2 which depicts the $1$-Wasserstein distance vs. layer index for the models included in our functional expert output analysis and provided a more nuanced interpretation of the PCA analysis that relates directly to our analysis in Section 3 and the overall expert granularity of the architecture.
* We enhance the cohesion between sections 3 and 4 by noting that REAP explicitly minimizes the reconstruction error bound and no longer make any assumptions with respect to the magnitude of the pruned gate-value, $g_j$.
* We added Figure 4c), which depicts the Jensen-Shannon divergence of compressed model logits vs. baseline logits as a function of completion token position. This figure shows that the merged model outputs diverge from baseline more rapidly than REAP pruned models.
* In our experimental results, we now highlight the effect of expert granularity. Low-granularity models, such as Llama-4-Scout and Mixtral, have relatively fewer experts per layer and active experts per token. Our analysis in Section 3 and empirical results in Section 5 suggest that low-granularity models may be more amenable to expert merging due to decreased router policy variance.
* Added Table A4 which tabulates the cumulative variance explained by PC1+PC2 in our PCA plots.
* Added Table A8 which tabulates Qwen3-480B-A35B-Coder-FP8 REAP pruned vs. baseline accuracy on $\tau^2$-bench.

---

> ### Author Response · Authors · 2025-11-27
> **Updated paper with highlighted changes**
>
> Our revised paper now includes blue font to differentiate text added since the original submission.

---

### Meta-Review · Area_Chair_Bja5 · 2026-01-06

**Summary:**

This paper studies one-shot MoE compression and argues that, for generative tasks, expert pruning is more reliable than expert merging. It proposes REAP (router-weighted activation pruning) and supports its claims with broad experiments across multiple SMoE families and benchmarks.

**Reviewer Concerns:**

Addressed by rebuttal:

+Reframes Sections 3–4 as intuition under assumptions (not universal proofs) and commits to clearer scoping.

+Strengthens “collapse” evidence with quantitative diagnostics (PCA variance explained, Wasserstein distance, JSD/logit divergence).

+Adds a merge+SFT experiment: improves merging but still below one-shot REAP in the reported setting.

+Provides clearer guidance on calibration domain sensitivity and adds quantization comparisons.


Still outstanding:

-Camera-ready must ensure the analysis avoids proof-like overreach (e.g., independence/factorization steps) and keeps claims tightly scoped.

-Merge+fine-tune results remain limited in scope (supportive but not exhaustive).

**Reviewer Scores:**

KnfQ: 6 → 7 (rebuttal answers practical concerns: calibration, post-merge tuning).
5xpD: 2 → 4 (maybe 5) (main objections—overstated theory + PCA-only evidence—are directly addressed with reframing + quantitative metrics).
Yea5: 8 → 8 (already positive; added comparisons/clarifications reinforce).

---

### Decision · Program_Chairs · 2026-01-26

Accept (Poster)